# Debiased Machine Learning and Network Cohesion for Doubly-Robust Differential Reward Models in Contextual Bandits

## Abstract

A common approach to learning mobile health (mHealth) intervention policies is linear Thompson sampling. Two desirable mHealth policy features are (1) pooling information across individuals and time and (2) incorporating a time-varying baseline reward. Previous approaches pooled information across individuals but not time, failing to capture trends in treatment effects over time. In addition, these approaches did not explicitly model the baseline reward, which limited the ability to precisely estimate the parameters in the differential reward model. In this paper, we propose a novel Thompson sampling algorithm, termed "DML-TS-NNR" that leverages (1) nearest-neighbors to efficiently pool information on the differential reward function across users *and* time and (2) the Double Machine Learning (DML) framework to explicitly model baseline rewards and stay agnostic to the supervised learning algorithms used. By explicitly modeling baseline rewards, we obtain smaller confidence sets for the differential reward parameters. We offer theoretical guarantees on the pseudo-regret, which are supported by empirical results. Importantly, the DML-TS-NNR algorithm demonstrates robustness to potential misspecifications in the baseline reward model.

## 1 Introduction

Mobile health (mHealth) and contextual bandit algorithms share a connection in the realm of personalized healthcare interventions. mHealth leverages mobile devices to deliver health-related services for real-time monitoring and intervention. Contextual bandit algorithms, on the other hand, are a class of machine learning techniques designed to optimize decision-making in situations where actions have contextual dependencies. The synergy arises when mHealth applications deploy contextual bandit algorithms to tailor interventions based on individual health data and context. For example, in a mobile health setting, a contextual bandit algorithm might dynamically adapt the type and timing of health-related notifications or interventions based on the user's current health status, historical behavior, and contextual factors like location or time of day Tewari & Murphy (2017).

At each decision point, a learner receives a context, chooses an action, and observes a reward. The goal is to maximize the expected cumulative reward. High-quality bandit algorithms achieve rewards comparable to those of an optimal policy. To achieve near-optimal performance in mobile health, bandit algorithms must account for (1) the time-varying nature of the outcome variable, (2) nonlinear relationships between states and outcomes, (3) the potential for intervention efficacy to change over time (due, for instance, to habituation as in Psihogios et al. (2019)), and (4) the fact that similar participants tend to respond similarly to interventions (Künzler et al., 2019).

Traditional mHealth intervention development—including just-in-time adaptive interventions (JITAIs), which aim to tailor the timing and content of notifications to maximize treatment effect (Nahum-Shani et al., 2018)—has centered on treatment policies pre-defined at baseline (e.g., Battalio et al. (2021); Nahum-Shani et al. (2021); Bidargaddi et al. (2018); Klasnja et al. (2019)). As the development of JITAIs shifts towards online learning (e.g., Trella et al. (2022); Liao et al. (2020); Aguilera et al. (2020)), we have the opportunity to incorporate the four key characteristics listed above into the development of optimal treatment policies through algorithms such as contextual bandits.

Although some solutions to these problems have been presented, no existing method offers a comprehensive solution that simultaneously addresses all four challenges in a satisfactory manner. The purpose of this paper is to fill this gap with a method that performs well in the mHealth setting where data is high-dimensional, highly structured, and often exhibits complex nonlinear relationships. To that end, this paper offers three main contributions: (1) A novel algorithm, termed as "DML-TS-NNR" that flexibly models the baseline reward via the double machine learning (DML) framework and pools efficiently across both users and time via nearest-neighbor regularization; (2) theoretical results showing that DML-TS-NNR achieves reduced confidence set sizes and an improved regret bound relative to existing methods; and (3) empirical analysis demonstrating the superior performance of DML-TS-NNR relative to existing methods in simulation and two recent mHealth studies.

The paper proceeds as follows. Section 2 summarizes related work. Section 3 describes the model and problem statement. Section 4 describes the algorithm along with the resulting theoretical results. Section 5 describes experimental results for simulations and two mobile health studies. Section 6 concludes with a discussion of limitations and future work.

## 2 RELATED WORK

The closest works are Choi et al. (2022) and Tomkins et al. (2021). Choi et al. (2022) employs a semi-parametric reward model for individual users and a penalty term based on the random-walk normalized graph Laplacian. However, limited information is provided regarding the explicit estimation of baseline rewards and the pooling of information across time. In contrast, Tomkins et al. (2021) carefully handles the issue of pooling information across users and time in longitudinal settings, but their approach (intelligentpooling) requires the baseline rewards to be linear and does not leverage network information. Below, we provide a summary of other relevant work in this area.

**Thompson Sampling.** Abeille & Lazaric (2017) showed that Thompson Sampling (TS) can be posed as a generic randomized algorithm constructed on the regularized least-squares (RLS) estimate rather than one sampling from a Bayesian posterior. At each step $t$, TS samples a perturbed parameter, where the additive perturbation is distributed so that TS explores enough (anti-concentration) but not too much (concentration). Any distribution satisfying these two conditions introduces the right amount of randomness to achieve the desired regret without actually satisfying any Bayesian assumption. We use the high-level proof strategy of Abeille & Lazaric (2017) in this work to derive our regret bound, although we need additional tools to handle our longitudinal setting with baseline rewards.

**Partially-linear bandits.** Greenewald et al. (2017) introduced a linear contextual bandit with a time-varying baseline and a TS algorithm with $\tilde{O}(d^2\sqrt{T})$ regret, where they used the inverse propensity-weighted observed reward as a pseudo-reward. By explicitly modeling the baseline, we obtain a pseudo-reward with lower variance. Krishnamurthy et al. (2018) improved this to $\tilde{O}(d\sqrt{T})$ regret using a centered RLS estimator, eliminating sub-optimal actions, and choosing a feasible distribution over actions. Kim & Paik (2019) proposed a less restrictive, easier to implement, and faster algorithm with a tight regret upper bound. Our regret bound (see Section 4) involves similar rates but is based on a different asymptotic regime that is not directly comparable due to the presence of an increasing pool of individuals.

**Nonlinear bandits.** (Li et al., 2017; Wang et al., 2019; Kveton et al., 2020) discussed generalized linear contextual bandit algorithms that accommodate nonlinear relationships via parametric link functions in a similar fashion to generalized linear models Nelder & Wedderburn (1972); McCullagh (2019). Other work (e.g., Snoek et al. (2015); Riquelme et al. (2018); Zhang et al. (2019); Wang & Zhou (2020)) allowed non-parametric relationships in both the baseline reward model *and* advantage function via deep neural networks; however, these approaches typically lack strong theoretical guarantees and are not designed for longitudinal settings in which pooling offers substantial benefit.

**Graph bandits.** In the study conducted by Cesa-Bianchi et al. (2013), individual-specific linear models were employed, accompanied by a combinatorial Laplacian penalty to encourage similarity among users' learned models. This approach yielded a regret bound of $\tilde{O}(nd\sqrt{T})$. Building upon this work, Yang et al. (2020) made further improvements by utilizing a penalty involving the random walk graph Laplacian. Their approach offers the following benefits: (1) it achieves a regret bound of $\tilde{O}(\Psi d\sqrt{nT})$ for some $\Psi \in (0,1)$ and (2) it reduces computational complexity from quadratic to linear by utilizing a first-order approximation to matrix inversion.

**Double Machine Learning.** Chernozhukov et al. (2018) introduced the DML framework, which provides a general approach to obtain $\sqrt{n}$-consistency for a low-dimensional parameter of interest in the presence of a high-dimensional or "highly complex" nuisance parameter. This framework combines Neyman orthogonality and cross-fitting techniques, ensuring that the estimator is insensitive to the regularization bias produced by the machine learning model. Moreover, it allows us to stay agnostic towards the specific machine learning algorithm while considering the asymptotic properties of the estimator. Later, a number of meta-learner algorithms were developed to leverage the DML framework and provide more precise and robust estimators (Hill, 2011; Semenova & Chernozhukov, 2021; Künzel et al., 2019; Nie & Wager, 2021; Kennedy, 2020).

**Doubly Robust Bandits.** Kim et al. (2021; 2023) use doubly robust estimators for contextual bandits in both the linear and generalized linear settings, respectively. They use them to obtain a novel regret bound with improved dependence on the dimensionality. In our setting, we use a doubly robust pseudo-reward (robust to either the propensity weights or the mean reward estimate being incorrect) in order to debias explicitly modeling the mean reward. We leave combining our approach with theirs for improved dependence on the dimensionality as future work.

## 3 Model and Problem Statement

We consider a doubly-indexed contextual bandit with a control action ($a = 0$) and $K$ non-baseline arms corresponding to different actions or treatments. Individuals $i = 1, 2, \ldots$ enter sequentially with each individual observed at a sequence of decision points $t = 1, 2, \ldots$. For each individual $i$ at time $t$, a context vector $S_{i,t} \in \mathcal{S}$ is observed, an action $A_{i,t} \in \{0, \ldots, K\} := [K]$ is chosen, and a reward $R_{i,t} \in \mathbb{R}$ is observed. In this paper, we assume the conditional model for the observed reward given state and action, i.e., $\mathbb{E}[R_{i,t}|S_{i,t} = s, A_{i,t} = a] := r_{i,t}(s, a)$, is given by

$$r_{i,t}(s, a) = \boldsymbol{x}(s, a)^\top \theta_{i,t} \delta_{a>0} + g_t(s), \tag{1}$$

where $\boldsymbol{x}(s, a) \in \mathbb{R}^{p \times 1}$ is a vector of features of the state and action, $\delta_{a>0}$ is an indicator that takes value 1 if $a > 0$ and 0 otherwise, and $g_t(s)$ is a baseline reward that observed when individuals are randomized to not receive any treatment. This can be an arbitrary, potentially nonlinear function of state $s$ and time $t$. Equation (1) is equivalent to assuming a linear *differential reward* for any $a > 0$; i.e., $\Delta_{i,t}(s, a) := r_{i,t}(s, a) - r_{i,t}(s, 0)$ is linear in $\boldsymbol{x}(s, a)$, whose parameter $\theta_{i,t} \in \mathbb{R}^p$ is allowed to depend both on the individual $i$ and time $t$.

To mimic real-world recruitment where individuals may not enter a study all at once, we consider a study that proceeds in *stages*. Figure 3 in Appendix A visualizes this sequential recruitment. At stage 1, the first individual is recruited and observed at time $t = 1$. At stage $k$, individuals $j \leq k$ have been observed for $k - j + 1$ decision times respectively. Then each individual $j \in [k + 1]$ is observed in a random order at their next time step. Let $\mathcal{H}_{i,t}$ denote the observation history up to time $t$ for individual $i$.

We make the following two standard assumptions as in Abeille & Lazaric (2017).

**Assumption 1.** *The reward is observed with additive error $\epsilon_{i,t}$, conditionally mean 0 (i.e., $\mathbb{E}[\epsilon_{i,t}|\mathcal{H}_{i,t}] = 0$) sub-Gaussian with variance $\sigma^2$: $\mathbb{E}[\exp(\eta\epsilon_{i,t})|\mathcal{H}_{i,t}] \leq \exp(\eta^2\sigma^2/2)$ for $\eta > 0$.*

**Assumption 2.** *We assume $\|\boldsymbol{x}(s, a)\| \leq 1$ for all contexts and actions and that there exists $B \in \mathbb{R}^+$ such that $\|\theta_{i,t}\| \leq B$, $\forall i, t$ and $|g_t(s)| \leq B$ $\forall s, t$ and $B$ is known.*

Here we consider *stochastic policies* $\pi_{i,t} : \mathcal{H}_{i,t} \times \mathcal{S} \to \mathcal{P}([K])$, which map the observed history $\mathcal{H}_{i,t}$ and current context to a distribution over actions $[K]$. Let $\pi_{i,t}(a|s)$ denote the probability of action $a \in [K]$ given current context $s \in \mathcal{S}$ induced by the map $\pi_{i,t}$ for a fixed (implicit) history.

### 3.1 DML and Doubly Robust Differential Reward

We first consider a single individual $i$ under a time-invariant linear differential reward, so that $\theta_{i,t} = \theta \in \mathbb{R}^p$. If the differential reward $\Delta(s_{i,t}, a_{i,t})$ was observed, we could apply ridge regression with a linear model of the form $\boldsymbol{x}(s_{i,t}, a_{i,t})^\top \theta$ and a ridge penalty of $\lambda\|\theta\|_2^2$. However, the differential reward is unobserved: we instead consider an inverse-probability weighted (IPW) estimator of the

differential reward based on the available data:

$$\mathbb{E}\left[\left(\frac{\delta_{A_{i,t}=\bar{a}}}{1-\pi_{i,t}(0|s)}-\frac{\delta_{A_{i,t}=0}}{\pi_{i,t}(0|s)}\right)R_{i,t}|s_{i,t},\bar{a}_{i,t}\right]=\Delta_{i,t}(s_{i,t},\bar{a}_{i,t}) \tag{2}$$

where $\bar{a}_{i,t}\in[K]$ denotes the *potential* non-baseline arm that may be chosen if the baseline arm is not chosen; i.e., randomization is restricted to be between $A_{i,t}=\bar{a}_{i,t}$ and $0$. Given the probabilities in the denominators are known, the estimator is unbiased and therefore can replace the observed reward in the Thompson sampling framework.

We refer to these $\Delta_{i,t}(s_{i,t},\bar{a}_{i,t})$ as the differential reward. Below, we define a *pseudo-reward* with the same expectation in reference to pseudo-outcomes from the causal inference literature (Bang & Robins, 2005; Kennedy, 2020). Let $f_{i,t}(s,a)$ be a working model for the true conditional mean $r_{i,t}(s,a)$. Then, following connections to pseudo-outcomes and doubly-robust (DR) estimators (Kennedy, 2020; Shi & Dempsey, 2023), we define the pseudo-reward $\tilde{R}_{i,t}^f(s,\bar{a})$ given state $S_{i,t}=s$ and potential arm $\bar{a}$

$$\tilde{R}_{i,t}^f(s,\bar{a})\equiv\frac{(R_{i,t}-f_{i,t}(s,A_{i,t}))}{\delta_{A_{i,t}=\bar{a}}-\pi_{i,t}(0|s)}+\Delta_{i,t}^f(s,\bar{a}) \tag{3}$$

where $\Delta_{i,t}^f(s,\bar{a})=f_{i,t}(s,\bar{a})-f_{i,t}(s,0)$. Going forward we will often abbreviate using $\tilde{R}_{i,t}^f$, with the state and action implied. Equation (3) presents a **D**oubly **R**obust estimator for the **D**ifferential **R**eward; i.e., if either $\pi_{i,t}$ or $f_{i,t}$ are correctly specified, (3) is a consistent estimator of the differential reward—so we refer to it as a *DR$^2$ bandit*. See Appendix G.1 for proof of double robustness. The primary advantage of this pseudo-reward is that by including the $f_{i,t}$, it has lower variance than if we simply used the inverse propensity-weighted observed reward as our pseudo-reward, which was done in Greenewald et al. (2017). Lemma 5 and Remark 2 in Appendix G.2 show proofs and discuss why this pseudo-reward lowers variance compared to Greenewald et al. (2017).

After exploring the properties of the pseudo-reward, an important question arises regarding how we can learn the function $f(s,a)$ using observed data. We hereby provide two options, each based on different assumptions. **Option 1** utilizes supervised learning methods and cross-fitting to accurately learn the function while avoiding overfitting as demonstrated in Chernozhukov et al. (2018) and Kennedy (2020). Our model (2) admits sample splitting across time under the assumption of additive i.i.d. errors and no delayed or spill-over effects. Such an assumption is plausible in the mHealth setting where we do not expect an adversarial environment.

In the following, we explain sample splitting as a function of time $t$ as we currently consider a single individual $i$. **Step 1**: Randomly assign each time $t$ to one of $M$-folds. Let $I_m(t)\subseteq\{1,\ldots,t\}$ denote the $m$-th fold as assigned up to time $t$ and $I_m^{\complement}(t)$ denote its complement. **Step 2**: For each fold at each time $t$, use any supervised learning algorithm to estimate the working model for $r_{i,t}(s,a)$ denoted $\hat{f}_{i,t}^{(m)}(s,a)$ using $I_m^{\complement}$. **Step 3**: Construct the pseudo-outcomes using (3) and perform weighted, penalized regression estimation by minimizing the loss function:

$$\sum_{m=1}^{M}\sum_{t\in I_m(T)}\tilde{\sigma}_{i,t}^2\left(\tilde{R}_{i,t}^{\hat{f}^{(m)}}-\boldsymbol{x}(s_{i,t},a_{i,t})^\top\theta\right)^2 \tag{4}$$

with ridge penalty $\lambda\|\theta\|_2^2$, where $\tilde{\sigma}_{i,t}^2=\pi_{i,t}(0|s_{i,t})\cdot(1-\pi_{i,t}(0|s_{i,t}))$. The weights are a consequence of unequal variances due to the use of DR estimators; i.e., $\text{var}(\tilde{R}_{i,t}^f)$ is inversely proportional to $(\tilde{\sigma}_{i,t}^2)^2$.

We explore an alternative, **Option 2**, based on recent work that avoids sample splitting via the use of stable estimators Chen et al. (2022). To relax the i.i.d. error assumption to Assumption 1, we only update $f_{i,t}(s,a)$ using observed history data in an online fashion, fixing pseudo-outcomes at each stage based on the current estimate of the nonlinear baseline. See Appendix C for further discussion.

Finally, in order to obtain guarantees for this DML approach, we make the following two assumptions: the first is on the convergence in $L^2$ (with expectation over states and actions) of our estimate $f$ to the true mean reward and boundedness of the estimator. Similar assumptions were made in Chen et al. (2022). The second is that the weights are bounded below, which would be a consequence if the

probability of taking no action is bounded above and below, an assumption made in Greenewald et al. (2017) .

**Assumption 3.** *Both* $\mathbb{E}_{p(s,\bar{a})}\left[(r_{i,t}(s,\bar{a}) - f_{i,t}(s,\bar{a}))^2\right] = o_P(k^{-1/2})$ *and* $\mathbb{E}_{p(s)}\left[(r_{i,t}(s,0) - f_{i,t}(s,0))^2\right] = o_P(k^{-1/2})$. *Further,* $|\hat{f}_{i,t}| \leq 2B$.

**Assumption 4.** *There exists* $c > 0$ *such that* $\tilde{\sigma}_{i,t}^2 > c$ *for all* $i, t$.

## 3.2 NEAREST NEIGHBOR REGULARIZATION

Above, we considered a single individual $i$ under a time-invariant linear differential reward function; i.e., $\boldsymbol{x}(s_{i,t}, a_{i,t})^\top \theta$ where $\theta \in \mathbb{R}^p$. Here, we consider the setting of $N$ independent individuals and a time-invariant linear differential reward with individual-specific parameter; i.e., $\theta_i \in \mathbb{R}^p$. If $\theta_i$ were known a priori, then one could construct a network based on $L_2$-distances $\{d(i,j) := \|\theta_i - \theta_j\|_2^2\}_{j \neq i}$.

Specifically, define a graph $G = (V, E)$ where each user represents a node, e.g., $V := [N]$, and $(i,j)$ is in the edge set $E$ for the smallest $M \ll N$ distances. The working assumption is that connected users share similar underlying vectors $\theta_i$, implying that the rewards received from one user can provide valuable insights into the behavior of other connected users. Mathematically, $(i,j) \in E$ implies that $\|\theta_i - \theta_j\|$ is small.

We define the Laplacian via the $N \times MN$ incidence matrix $B$. The element $B_{v,e}$ corresponds to the $v$-th vertex (user) and $e$-th edge. Denote the vertices of $e$ as $v_i$ and $v_j$ with $i \, \text{¿} \, j$. $B_{ve}$ is then equal to 1 if $v = v_i$, -1 if $v = v_j$, and 0 otherwise. The Laplacian matrix is then defined as $L = BB^\top$. We can then adapt (4) by summing over participants and including a *network cohesion* penalty similar to Yang et al. (2020):

$$\text{tr}(\Theta^\top L \Theta) = \sum_{(i,j) \in E} \|\theta_i - \theta_j\|_2^2,$$

where $\Theta := (\theta_1, \ldots, \theta_N)^\top \in \mathbb{R}^{N \times p}$. The penalty is small when $\theta_i$ and $\theta_j$ are close for connected users. Following Assumption 2 and above discussion, we further assume:

**Assumption 5.** *There exists* $D \in \mathbb{R}^+$ *such that* $\|\theta_i - \theta_j\|_2^2 \leq D$, $\forall i, j$, *and* $D$ *is known.*

# 4 DML THOMPSON SAMPLING WITH NEAREST NEIGHBOR REGULARIZATION

## 4.1 ALGORITHM

Based on Section 3, we can now formally state our proposed DML **T**hompson **S**ampling with **N**earest **N**eighbor **R**egularization (DML-TS-NNR) algorithm. In our study, we adopt a sequential recruitment setting in which individuals' enrollment occurs in a staggered manner to mimic the recruitment process in real mHealth studies. More specifically, we first observe individual $i = 1$ at time $t = 1$. Then we observe individuals $i = (1, 2)$ at times $t = (2, 1)$. After $k$ time steps, we observe individuals $i \in [k]$ at times $(k + 1 - i, k - i, \ldots, 1)$ respectively. We then observe these individuals in a random sequence one at a time before moving to stage $k + 1$. Define $\mathcal{O}_k = \{(i, t) : i \leq k \,\&\, t \leq k + 1 - i\}$ be the set of observed time points across all individuals at stage $k$. Again see Figure 3 in Appendix A for a visualization.

By performing a joint asymptotic analysis with respect to the total number of individuals ($N$) and time points ($T$), we can relax the assumption of a single time-invariant linear advantage function and allow $\theta_{i,t} \in \mathbb{R}^p$ to depend on both the individual $i$ and time $t$. Here, we let $\theta_{i,t} = \theta + \theta_i^{\text{ind}} + \theta_t^{\text{time}}$; i.e., include (i) an individual-specific, time-invariant term $\theta_i$, and (ii) a shared, time-specific term $\theta_t$. This setup is similar to the intelligentpooling method of Tomkins et al. (2021); however, rather than assume individuals and time points are unrelated iid samples, we assume knowledge of some network information (e.g., the similarity of certain individuals or proximity in time) and regularize these parameters accordingly to ensure network cohesion.

The DML-TS-NNR algorithm is shown in Algorithm 1. To see the motivation, consider the following. We first assume that we have access to two nearest neighbor graphs, $G_{\text{user}}$ and $G_{\text{time}}$, where each characterizes proximity in the user- and time-domains respectively. Then at at stage $k$, we estimate

all parameters, e.g. $\Theta_k = \text{vec}[(\theta, \theta_1^{\text{ind}}, \ldots, \theta_k^{\text{ind}}, \theta_1^{\text{time}}, \ldots, \theta_k^{\text{time}})] \in \mathbb{R}^{p(2k+1)}$, by minimizing the following penalized loss function $L_k(\Theta_k; \lambda, \gamma)$, which is defined as the following expression:

$$
\sum_{(i,t)\in\mathcal{O}_k} \tilde{\sigma}_{i,t}^2 \left( \tilde{R}_{i,t}^{\hat{f}^{(m)}} - \boldsymbol{x}(S_{i,t}, A_{i,t})^\top (\theta + \theta_i^{\text{ind}} + \theta_t^{\text{time}}) \right)^2 +
$$

$$
\gamma \left( \|\theta\|_2^2 + \sum_{i=1}^{k} \|\theta_i^{\text{ind}}\|_2^2 + \sum_{t=1}^{k} \|\theta_t^{\text{time}}\|_2^2 \right) + \lambda \left( \text{tr} \left( \Theta_{\text{user}}^\top L_{\text{user}} \Theta_{\text{user}} \right) + \text{tr} \left( \Theta_{\text{time}}^\top L_{\text{time}} \Theta_{\text{time}} \right) \right),
$$

(5)

where $\Theta_{\text{user}}, \Theta_{\text{time}} \in \mathbb{R}^{p\times k}$. In comparison to existing methods, the primary novelty in Equation (5) is that (1) the observed outcome $R_{i,t}$ is replaced by a pseudo-outcome $\tilde{R}_{i,t}^{f^{(k)}}$ and (2) the doubly-robust pseudo-outcome leads to a weighted least-squares loss with weights $\tilde{\sigma}_{i,t}^2$. The network cohesion penalties and time-specific parameters have been considered elsewhere (Yang et al., 2020; Tomkins et al., 2021), though, not together. For more details regarding Algorithm 1, please refer to Appendix B.

---

**Algorithm 1** DML-TS with Nearest Neighbor Regularization (DML-TS-NNR)

---

**Input**: $\delta, \sigma, c, C, m, \lambda, \gamma, L, B_w, D_w$
**Set** $L_\otimes = L \otimes I_p$ and $B = k\frac{\lambda}{\sqrt{\gamma}}(D_{\text{ind}} + D_{\text{time}}) + \sqrt{\gamma n}(B_{\text{ind}} + B_{\text{time}})$
**Initialize:** $V_0 = \text{diag}(\gamma I_p, \lambda L_\otimes^{\text{ind}} + \gamma I_{kp}, \lambda L_\otimes^{\text{time}} + \gamma I_{kp})$ and $b_0 = \mathbf{0}$
**for** $k = 1, \ldots, K$ **do**
  **Option 1:** Randomly assign $(i,t) \in \mathcal{O}_k \backslash \mathcal{O}_{k-1}$ to one of the $M$ partitions
  Observe Context variable $S_l = S_{i_l, k+i_l-1}$
  Set $\hat{\Theta}_k = V_k^{-1} b_k$
  Calculate
$$
\beta_k(\delta) = v_k \left[ 2\log\left( \frac{\det(V_k)^{1/2}}{\det(V_0)^{1/2}\delta/2} \right) \right]^{1/2} + B
$$
  where $v_k^2 \equiv Cc\log^{2m}(k)k^{-1/2} + \sigma^2 c^2$
  Generate $\eta_k \sim \mathcal{D}^{TS}$ and compute
$$
\tilde{\Theta}_k = \hat{\Theta}_k + \beta_k(\delta')V_k^{-1/2}\eta_k
$$
  For each $(i,t) \in \mathcal{O}_k \backslash \mathcal{O}_{k-1}$ select $A_{i,t}$ that maximizes:
$$
\boldsymbol{x}(S_{i,t}, a)^\top \left( \tilde{\theta} + \tilde{\theta}_i^{\text{ind}} + \tilde{\theta}_t^{\text{time}} \right)
$$
  Observe rewards $R_{i,t}$
  Construct feature $\boldsymbol{x}_t = \boldsymbol{x}(S_t, A_t)$ and $\phi_{i,t} = \phi(\boldsymbol{x}_{i,t})$
  **Option 1:** Re-construct predictions for all $\hat{f}^{(m)}$ partitions for $m = 1, \ldots, M$ and re-compute all pseudo-outcomes $\tilde{R}_{i,t}^{\hat{f}^{(m)}}$ for all $(i,t) \in \mathcal{O}_k$.
  **Option 2:** Construct predictions for next stage $\hat{f}^{(k)}$ partitions and compute pseudo-outcomes $\tilde{R}_{i,t}^{\hat{f}^{(k)}}$ only for those $(i,t) \in \mathcal{O}_k \backslash \mathcal{O}_{k-1}$.
  Update $V_k = V_{k-1} + \sum_{(i,t)\in\mathcal{O}_k\backslash\mathcal{O}_{k-1}} \tilde{\sigma}_{i,t}^2 \phi_{i,t}\phi_{i,t}^\top$ and $b_k = b_{k-1} + \sum_{(i,t)\in\mathcal{O}_k\backslash\mathcal{O}_{k-1}} \tilde{\sigma}_{i,t}^2 \tilde{R}_{i,t}^{\hat{f}} \phi_{i,t}$
**end for**

---

## 4.2 Regret Analysis

Given the knowledge of true parameters $\Theta$, the optimal policy is simply to select, at decision time $t$ for individual $i$, the action $a_{i,t}^* = \arg\max_{a\in\mathcal{A}} \boldsymbol{x}(S_{i,t}, a)^\top (\theta + \theta_i + \theta_t)$ given the state variable $S_{i,t}$. This leads us to evaluate the algorithm by comparing it to this optimal policy after each stage. Given both the number of individuals and the number of time points increases per stage, we define stage $k$

regret to be the average across all individuals at stage $k$:

$$\textbf{Regret}_K = \sum_{k=1}^{K} \frac{1}{k} \left\{ \sum_{(i,t)\in\mathcal{O}_k\setminus\mathcal{O}_{k-1}} \left[ \boldsymbol{x}(S_{i,t}, a_{i,t}^*)^\top (\theta^* + \theta_i^\star + \theta_t^\star) - \boldsymbol{x}(S_{i,t}, A_{i,t})^\top (\theta^* + \theta_i^\star + \theta_t^\star) \right] \right\}$$

This is a version of pseudo-regret Audibert et al. (2003). Compared to standard regret, the randomness of the pseudo-regret is due to $\{A_{i,t}\}_{t=1}^\top$ since the error terms $\{\epsilon_{i,t}\}_{t=1}^\top$ are removed in the definition.

**Theorem 1.** *Under Assumptions 1 and 2, with probability at least $1 - \delta$, Algorithm 1's regret satisfies*

$$\left( \beta_K(\delta') + \gamma_K(\delta') \left[ 1 + \frac{4}{d} \right] \right) \sqrt{4cH_K Kd \log\left( \gamma + \lambda M + \frac{K+1}{8d} \right) - \log\det(V_0)}$$

$$+ \frac{4\gamma_K(\delta')}{d} \sqrt{\frac{8K}{\lambda} \log\left( \frac{4}{\delta} \right)},$$

*where $f_K = \log\det V_K - \log\det V_0$, $V_0 = \lambda L \otimes I_p + \gamma I_{np}$, $\delta' = \delta/4K$, $\min(\pi(0|s), 1 - \pi(0|s)) > 1/c$ and $H_K = O(\log(K))$ is the harmonic number. $\beta_K$ and $\gamma_K$ are defined in Appendix G.*

Proof of Theorem 1 is in Appendix G. The regret bound is similar to prior work by Abeille & Lazaric (2017); however, our bound differs in three ways: (1) the harmonic number $H_K$ enters as an additional cost for considering average regret per stage with the regret being $O(\sqrt{K}\log^2(K))$ having an additional $\log(K)$ factor; (2) the bound depends on the dimension of the differential reward model rather than the dimension of the overall model which can significantly improve the regret bound; and (3) the main benefit of our use of DML is in $\beta_K(\delta')$ and $\gamma_K(\delta')$, which depend on the rate of convergence of the model $f$ to the true mean differential reward $r$ as discussed in Appendix G.2, which demonstrates the benefits of good models for this term and how it impacts regret.

Note this regret bound is sublinear in the number of stages and scales only with the differential reward complexity $d$, not the complexity of the baseline reward $g$. As the second term scales with $\sqrt{K}$, for large $K$ we can focus on the first term which scales $O\left( \sqrt{c \cdot d \cdot \log^2(K)K} \right)$. Prior work scales sublinearly with the number of decision times $T$ as they assume either a single contextual bandit ($n = 1$) or a fixed number of individuals $n$. Greenewald et al. (2017) scales as $O(d^2\sqrt{T\log(T)})$ while Yang et al. (2020) scale as $O(\sqrt{\tilde{d}nT\log(T)})$ where $\tilde{d}$ is the complexity of the joint baseline and differential reward model. Interestingly, at stage $K$ we see $n = K$ individuals over $n = K$ decision times (with different number of observations per individual); however, we do not see a regret scale with $\sqrt{nT} = K$. Instead, we only receive an extra $\log(K)$ factor reflecting the benefit of pooling on average regret.

There are several technical challenges to this regret bound. First, the confidence ellipsoids $\beta_k$ and $\gamma_k$ depend on the sub-Gaussian variance factor of the pseudo-reward and need to be derived for the DML pseudo-reward. Second, two results in Abbasi-Yadkori et al. (2011) need to be reproven: the first is Lemma 7, a linear predictor bound that is used as a key step to derive the final regret bound. It requires handling the fact that our regularized least squares estimate now uses our DML pseudo-reward instead of the observed reward. The second, Proposition 1, requires care as we are now doing *weighted* regularized least squares. The original version used in Abeille & Lazaric (2017) requires an upper bound on the sum of squared feature norms, but applying Abbasi-Yadkori et al. (2011) only gives us an upper bound on the sum of weighted squared norms. In order to derive the needed bound, we use assumption 4 and then apply Abbasi-Yadkori et al. (2011) to obtain our version of their results that has an upper bound that depends on $c > 0$, the lower bound on the weights. Finally, the regret bound itself needs to handle the fact that we have stages with multiple individuals (increasing by one) per stage. This leads to a sum over stages and participants within stages of the difference between RLS and TS (and RLS and true) linear predictors. By some manipulation and an application of Cauchy Schwartz, we see a sum of $\frac{1}{k}$ over stages, which leads to the harmonic number, which describes the additional cost of handling multiple participants in a study in stages.

# 5 EXPERIMENTS

## 5.1 COMPETITOR COMPARISON SIMULATION

In this section, we test three versions of our proposed method: (1) DML-TS-NNR-BLM: Our algorithm using an ensemble of **B**agged **L**inear **M**odels, (2) DML-TS-NNR-BT: Our algorithm using an ensemble of **B**agged stochastic gradient **T**rees (Gouk et al., 2019; Mastelini et al., 2021), and (3) DML-TS-SU-BT: Same as (2) but treating the data as if it were derived from a **S**ingle **U**ser.

We implemented these using **Option 2** in Algorithm 1, and compared to four related methods: (1) Standard: Standard Thompson sampling for linear contextual bandits, (2) AC: **A**ction-**C**entered contextual bandit algorithm (Greenewald et al., 2017), (3) IntelPooling: The intelligentpooling method of Tomkins et al. (2021) fixing the variance parameters close to their true values, and (4) Neural-Linear: a method that uses a pre-trained neural network to transform the feature space for the baseline reward (similar to the Neural Linear method of Riquelme et al. (2018), which in turn was inspired by Snoek et al. (2015)). In general, we expect our method to outperform these methods because it is the only one that can (1) efficiently pool across users and time, (2) leverage network information, and (3) accurately model a complex, nonlinear baseline reward.

We compare these seven methods under three settings that we label as Homogeneous Users, Heterogeneous Users, and Nonlinear. The first two settings involve a linear baseline model and time-homogeneous parameters, but they differ in that the users in the second setting have distinct parameters. The third setting is more general and includes a nonlinear baseline, user-specific parameters, and time-specific parameters. Across all three settings, we simulate 125 stages following the staged recruitment regime depicted in Figure 3 in Appendix A, and we repeat the full 125-stage simulation 50 times. Appendix D provides details on the setup and a link to our implementation.

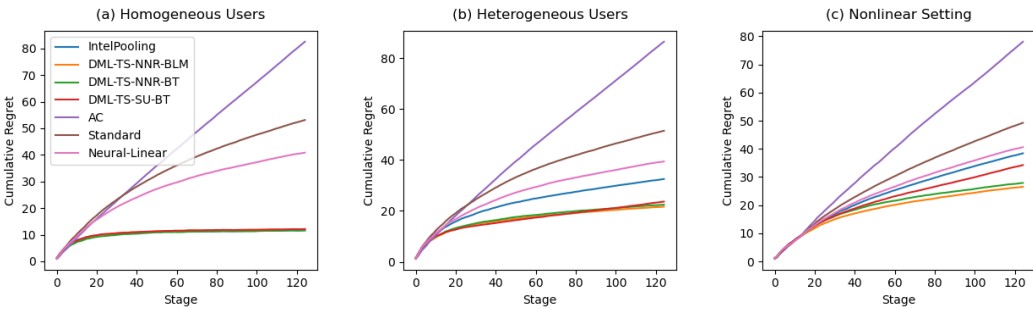

Figure 1: Cumulative regret in the (a) Homogeneous Users, (b) Heterogeneous Users, and (c) Nonlinear settings. The DML methods perform competitively in all three settings and appear to be achieving sublinear regret as expected based on our theoretical results. The DML-TS-NNR-BLM and DML-TS-NNR-BT algorithms perform best, and their final regret is statistically indistinguishable (see Table 1 in Appendix D.2).

Figure 1 shows the cumulative regret for each method at varying stages. DML methods perform competitively against the benchmark methods in all three settings and achieve sublinear regret as expected based on our theoretical results. Across all settings, the best-performing method is either DML-TS-NNR-BLM or DML-TS-NNR-BT. In the first setting, the difference between our methods and IntelPooling is not statistically meaningful because IntelPooling is properly specified and network information is not relevant. In the other two settings, our methods offer substantial and statistically meaningful improvement over the other methods. Appendices D.2 and D.3 shows detailed pairwise comparisons between methods and an additional simulation study using a rectangular array of data.

## 5.2 VALENTINE RESULTS

In parallel with the simulation study, we conducted a comparative analysis on a subset of participants from the Valentine Study (Jeganathan et al., 2022), a prospective, randomized-controlled, remotely-administered trial designed to evaluate an mHealth intervention to supplement cardiac rehabilitation

for low- and moderate-risk patients. In the analyzed subset, participants were randomized to receive or not receive contextually tailored notifications promoting low-level physical activity and exercise throughout the day. The six algorithms being compared include (1) Standard, (2) AC, (3) IntelPooling, (4) Neural-Linear, (5) DML-TS-SU-RF (RF stands for Random Forest (Breiman, 2001)), and (6) DML-TS-NNR-RF. Figure 2 shows the estimated improvement in average reward over the original constant randomization, averaged over stages (K = 120) and participants (N=108).

To demonstrate the advantage of our proposed algorithm in terms of average reward compared to the competing algorithms, we conducted a pairwise paired t-test with a one-sided alternative hypothesis. The null hypothesis ($H_0$) stated that two algorithms achieve the same average reward, while the alternative hypothesis ($H_1$) suggested that the column-indexed algorithm achieves a higher average reward than the row-indexed algorithm. Figure 2 displays the p-values obtained from these pairwise t-tests. Since the alternative hypothesis is one-sided, the resulting heatmap is not symmetric. More details on implementation can be found in Appendix E.

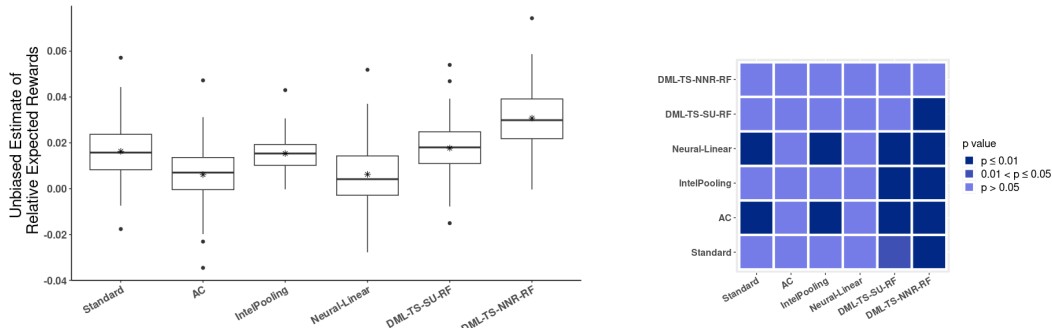

Figure 2: **(left)** Boxplot of unbiased estimates of the average per-trial reward for all six competing algorithms, relative to the reward obtained under the pre-specified Valentine randomization policy across 100 bootstrap samples. Within each box, the asterisk ($*$) indicates the mean value, while the mid-bar represents the median. **(right)** Heatmap of p-values from the pairwise paired t-tests. The last column's dark shade indicates that the proposed DML-TS-NNR-RF algorithm achieves significantly higher rewards than the other five competing algorithms. This implies that after implementing our proposed algorithm, the step counts increased by $3.5\%$ more than what the constant randomization policy achieved.

To further enhance the competitive performance of our proposed DML-TS-NNR algorithm, we conducted an additional comparative analysis using a real-world dataset from the Intern Health Study (IHS) (NeCamp et al., 2020). Further details regarding the analysis can be found in Appendix F.

## 6 Discussion and Future Work

In this paper, we have presented the DML Thompson Sampling with Nearest Neighbor Regularization (DML-TS-NNR) algorithm, a novel contextual bandit algorithm specifically tailored to the mHealth setting. By leveraging the DML framework and network cohesion penalties, DML-TS-NNR is able to accurately model complex, nonlinear baseline rewards and efficiently pool across both individuals *and* time. The end result is increased statistical precision and, consequently, the ability to learn effective, contextually-tailored mHealth intervention policies at an accelerated pace.

While DML-TS-NNR achieves superior performance relative to existing methods, we see several avenues for improvement. First, the algorithm considers only immediate rewards and, as such, may not adequately address the issue of treatment fatigue. Second, the current algorithm involves computing a log-determinant and matrix inverse, which can be computationally expensive for large matrices. Third, we have made the simplifying assumption that the differential reward is linear in the context vectors. Fourth, we have assumed that the network structure is known and contains only binary edges. Fifth, our algorithm involves several hyperparameters whose values may be difficult to specify in advance. Future work will aim to address these practical challenges in applied settings.

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
