# Debiased Machine Learning and Network Cohesion for Doubly-Robust Differential Reward Models in Contextual Bandits

## Abstract

A common approach to learning mobile health (mHealth) intervention policies is linear Thompson sampling. Two desirable features of an mHealth policy are (1) pooling information across individuals and time and (2) modeling the differential reward linear model with a time-varying baseline reward. Previous approaches focused on pooling information across individuals but not time, thereby

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

 assumption on the convergence of our estimate $\hat{f}$ to the true mean reward:

**Assumption 3.** *For some $m > 0$, at stage $k$, for any $\delta > 0$, there exists $C > 0$ s.t.*

$$P \left( \frac{\|\hat{f}_{i,t} - r_{i,t}\|_{\infty}}{k^{-1/4} \log^m k} > C \right) \leq \delta$$

*That is, $\|\hat{f}_{i,t} - r_{i,t}\|_\infty = \tilde{O}_P(k^{-1/4})$. Further, $\hat{f}_{i,t}$ is uniformly bounded $\forall i, t$ and stage $k$.*

Note that while $L^\infty$ convergence is strong, a number of machine learning models such as KNN (Jiang, 2019) are known to exhibit it stochastically. We leave relaxing this assumption to the weaker $L^2$ (mean squared) convergence assumption for future work.

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

# Appendices

## A  RECRUITMENT REGIME ILLUSTRATION

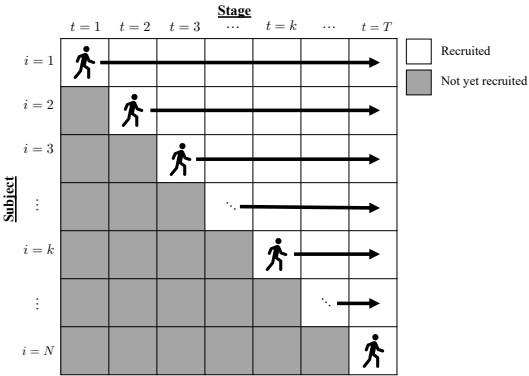

Figure 3: Illustration of the staged recruitment scheme. At each recruitment stage (each time point), a new participant is recruited and observed; at the same time, all participants who were recruited prior to the current stage are also observed again. Observations are not collected from participants who have yet to be recruited. For simplicity, we assume one participant is recruited at each stage.

## B  ADDITIONAL DETAILS FOR ALGORITHM 1

The inclusion of $\gamma > 0$ ensures a unique solution at each step, while $\lambda$ controls smoothness across the individual-specific and time-specific parameters. In particular, when $\lambda = 0$, no information is shared across individuals or across time. A positive value of $\lambda$ introduces the smoothness of these estimates, e.g., if $(i,j) \subset E_{\text{ind}}$ then $\|\theta_i^{\text{ind}} - \theta_j^{\text{ind}}\|_2$ tends to be small. It can be shown that $\sum_{(i,j)\subset E_{\text{ind}}} \|\theta_i^{\text{ind}} - \theta_j^{\text{ind}}\|_2^2 = \theta_{\text{ind}}^\top L_\otimes \theta_{\text{ind}}$, where $L_\otimes = L \otimes I_p$ and $\theta_{\text{ind}} = (\theta_1^{\text{ind}}, \ldots, \theta_k^{\text{ind}})$ is the vector of individual-specific parameters. Let $\boldsymbol{x}_{i,t} := \boldsymbol{x}(S_{i,t}, A_{i,t})$ and $V_0 = \text{diag}(\gamma I_p, \lambda L_\otimes^{\text{ind}} + \gamma I_{kp}, \lambda L_\otimes^{\text{time}} + \gamma I_{kp})$. Let $\Phi_k$ be the design matrix of features $\phi(\mathbf{x}_{i,t})$, $W_k$ be the diagonal matrix of weights $\tilde{\sigma}_{i,t}^2$, and $R_k$ the vector of pseudo-rewards $\tilde{R}_{i,t}^{f^{(k)}}$. In all of these $(i,t) \in \mathcal{O}_k$. Then the minimizer $\hat{\Theta}_k$ of (5) is

$$[\Phi_k^\top W_k \Phi_k + V_0]^{-1}[\Phi_k^\top W_k R_k] = \underbrace{\Big[ \sum_{(i,t)\in\mathcal{O}_k} \tilde{\sigma}_{i,t}^2 \phi(\boldsymbol{x}_{i,t})\phi(\boldsymbol{x}_{i,t})^\top + V_0 \Big]^{-1}}_{V_{k-1}^{-1}} \underbrace{\Big[ \sum_{(i,t)\in\mathcal{O}_k} \tilde{\sigma}_{i,t}^2 \tilde{R}_{i,t}^{f^{(k)}} \phi(\boldsymbol{x}_{i,t}) \Big]}_{b_{k-1}}.$$

where $\phi(\boldsymbol{x}_{i,t}) \in \mathbb{R}^{p(2k+1)}$ for any $(i,t) \in \mathcal{O}_k$ is defined as

$$\phi(\boldsymbol{x}_{i,t}) = (\boldsymbol{x}_{i,t}, \bar{\mathbf{0}}_{(i-1)p}, \boldsymbol{x}_{i,t}, \bar{\mathbf{0}}_{(k-i)p}, \bar{\mathbf{0}}_{(t-1)p}, \boldsymbol{x}_{i,t}, \bar{\mathbf{0}}_{(k-t)p}).$$

The first location of non-zero entries is for the global parameters, the second for the individual parameters and the third for the time parameters.

**Remark 1** (Computationally efficient estimation). *Direct calculation of $\hat{\Theta}_k$ leads to a computationally expensive inversion of the $(2k+1) \cdot p$ dimensional matrix $V_{t-1}$ at each stage $k$. To avoid this, we observe $V_k = V_{k-1} + \sum_{(i,t)\in\mathcal{O}_k \setminus \mathcal{O}_{k-1}} \tilde{\sigma}_{i,t}^2 \phi(\boldsymbol{x}_{i,t})\phi(\boldsymbol{x}_{i,t})^\top$ and apply the Sherman-Morrison formula for more efficient computation.*

Next, we consider a generic randomized algorithm based on the RLS estimate by sampling a perturbed parameter $\tilde{\Theta}_k$ and then selecting an action by simply maximizing the linear differential reward $\boldsymbol{x}(s,a)^\top(\tilde{\theta} + \tilde{\theta}_i + \tilde{\theta}_t)$. This construction includes standard Thompson sampling as an important special case. Specifically, we construct $\tilde{\Theta}_k = \hat{\Theta}_k + \beta_k(\delta')V_k^{-1/2}\eta_k$ where $\eta_k$ is a random sample drawn i.i.d. from a suitable multivariate distribution $\mathcal{D}^{\text{TS}}$ and $\beta_k(\delta')$ is a term from the self-normalization bound (Theorem 2) developed in Abbasi-Yadkori

et al. (2011). Because of our use of $f_{i,t}$ to approximate $r_{i,t}$, $\beta_k(\delta')$ is smaller, and thus our distribution for $\hat{\Theta}_k$ has lower variance than if we did not use $f_{i,t}$. We discuss this in detail in Appendix G.2. To ensure small regret, we must choose the random variable $\eta_k$ to ensure *sufficient exploration but not too much.* Definition 1 is from Abeille & Lazaric (2017) and formalizes the properties of the random variable $\eta_k$.

**Definition 1.** $\mathcal{D}^{TS}$ *is a multivariate distribution on $\mathbb{R}^d$ absolutely continuous with respect to Lebesgue measure which satisfies: 1. (anti-concentration) that there exists a strictly positive probability $p$ such that for any $u \in \mathbb{R}^d$ with $\|u\| = 1$, $\mathbb{P}(u^\top \eta \geq 1) \geq p$; and 2. (concentration) there exists $c$, $c'$ positive constants such that $\forall \delta \in (0,1)$, $P(\|\eta\| \leq \sqrt{cd\log(c'd/\delta)}) \geq 1 - \delta$.*

While a Gaussian prior satisfies Definition 1, this approach allows us to move beyond Bayesian posteriors to generic randomized policies. In practice, the true parameter values $\Theta$ are unknown, so $\beta_k(\delta')$ is not available. Thus one needs to insert upper bounds for $\|\mathcal{L}_\otimes^{\mathrm{ind}} \theta_{\mathrm{ind}}\|_2$ and $\|\theta_{\mathrm{ind}}\|_2$ and similarly for time-specific parameters. For example, $\|L_\otimes \theta\|_2 \leq M \cdot N \max_{(i,j) \in E} \|\theta_i - \theta_j\|_2$. Similarly, based on $\|\theta\|_2 \leq \sqrt{k} \max_i \|\theta_i\|_2$, by Assumption 2, we have $\|\theta\|_2 \leq \sqrt{k}B$.

## C    OPTION 1 & 2

In Algorithm 1, we provide two options for constructing $\hat{f}$ and the corresponding pseudo-outcomes $\tilde{R}_{i,t}^{\hat{f}}$. Option 1 assumes i.i.d. additive error and recomputes the pseudo-rewards, $\tilde{R}_{i,t}^{\hat{f}^{(m)}}$, at each stage for all $i, t$ using an updated estimate of $f$, which can be estimated using either (a) sample splitting or (b) stable estimators, such as bagged ensembles constructed via sub-sampling. Option 2 uses historical data to generate predictions in an online fashion, and calculates the pseudo-rewards only once without updating them for all subsequent stages.

Our simulations have confirmed that both options result in comparable regret. In reality, however, it's important to note that the decisions made for individual $i$ at time $t$ may depend on all previous data. Based on this argument, **Option 2** was used in *both* case studies and simulations, highlighting the benefits of our approach in numerical as well as real-world mHealth studies, as detailed in the manuscript.

### C.1    OPTION 1

Sample splitting has been widely used in the DML-related literature (Chernozhukov et al., 2018; Kennedy, 2020) to relax modeling constraints in constructing $\hat{f}$. This approach allows the flexibility of the model to increase with the sample size while protecting against overfitting. As a result, complex machine learning algorithms can be utilized to estimate the function $f$, resulting in accurate estimation of the differential reward $\Delta^f$ and, consequently, precise estimates of the $\theta$'s in the linear differential reward function.

When implemented as part of our proposed algorithm, sample splitting randomly partitions all available data into $M$ folds. This step relies heavily on the assumption of i.i.d additive errors, as the random splits are formed on a per-observation basis. Theoretical findings outlined in Chernozhukov et al. (2018) demonstrate the crucial role of sample splitting in achieving $\sqrt{n}$-consistency and ensuring the validity of inferential statements for the parameters in the linear differential reward function.

As an alternative to sample splitting, one can instead construct $\hat{f}$ via stable estimators that exhibit a $o(n^{-1/2})$ leave-one-out stability. A recent study by Chen et al. (2022) demonstrated that estimators based on predictive models that satisfy this condition can achieve $\sqrt{n}$-consistency and asymptotic normality without relying on the Donsker property or employing sample splitting.

The stability conditions outlined in Theorem 5 of Chen et al. (2022) are satisfied by bagging estimators formed with sub-sampling. We leverage this result in Section 5 by testing several versions of our method based on bagged estimators: two based on bagged stochastic gradient

trees (DML-TS-NNR-BT and DML-TS-SU-BT) and one based on bagged linear models (DML-TS-NNR-BLM).

As long as the errors are i.i.d., methods based on either (a) sample splitting or (b) stable estimators are permissible within our method. In both cases, we are able to leverage both current and previous observations to construct the latest estimation of $\hat{f}$.

### C.2 OPTION 2

In the mobile health setting, deviations from the expected reward typically represent the effects of idiosyncratic error as opposed to adversarial actions. Consequently, the i.i.d. assumption is more plausible in mobile health than in general contextual bandit settings. However, due to the sequential design of mobile health studies, one challenge that may arise is dependence across time within individual users. Option 2 adapts our method to address this challenge.

When considering errors that exhibit temporal dependence, it is reasonable to utilize past observations to create future pseudo-rewards but not the other way around. For instance, at stage $k$, we train the model $\hat{f}^{(k)}$ using all the history observed up until stage $k-1$ and construct pseudo-rewards for the rewards observed in stage $k$; however, we do not use $\hat{f}^{(k)}$ to update pseudo-rewards for previous stages.

Fixing the pseudo-rewards in this manner enables us to move beyond the assumption of i.i.d. additive errors. This approach is compatible with both the sample splitting and bagging approaches discussed previously. As a computational convenience, researchers may choose to update $\hat{f}$ in an online fashion as we did in Section 5.1. The primary tradeoff in doing so is that the online predictive model may not perform as well as a model trained in one batch, especially in the early stages.

## D ADDITIONAL DETAILS FOR SIMULATION STUDY

The code for the simulation study is fully containerized and publicly available at `https://redacted/for/anonymous/peer/review`.

### D.1 SETUP DETAILS

We consider a generative model of the following form for user $i$ at time $t$:

$$R_{it} = g(S_{it}) + x(S_{it}, A_{it})\,\theta_{it} + \epsilon_{it}, \quad \epsilon_{it} \sim \mathcal{N}(0, 1)$$

Here $S_{it} = (s_1, s_2) \in \mathbb{R}^2$ is a context vector, with both dimensions $\overset{iid}{\sim} U(-1, 1)$. We set $x(s, a) = a\,(1, s_1, s_2)$. For simplicity, we set $g$ to a time-homogeneous function. The specific nature of the function varies across the following three settings mentioned in Section 5.1:

- Homogeneous Users: Standard contextual bandit assumptions with a linear baseline and no user- or time-specific parameters. The linear baseline is $g(S_{it}) = 2 - 2s_1 + 3s_2$, and the causal parameter is $\theta_{it} = (1, 0.5, -4)$ such that the optimal action varies across the state space.

- Heterogeneous Users: Same as the above but each user's causal parameter has iid $\mathcal{N}(0, 1)$ noise added to it.

- Nonlinear: The general setting discussed in the paper with a nonlinear baseline, user-specific parameters, and time-specific parameters. The base causal parameter and user-specific parameters are the same as in the previous two settings. The nonlinear baseline and time-specific parameter are shown in Figure 4.

We assume that the data are observed via a staged recruitment scheme, as illustrated in Figure 3 in Appendix A. For computational convenience, we update parameters and

select actions in batches. If, for instance, we observe twenty users at a given stage, we update our estimates of the relevant causal parameters and select actions for all twenty users simultaneously. This strategy offers a slight computational advantage with limited implications in terms of statistical performance.

For simplicity, we assume that the nearest neighbor network is known and set the relevant hyperparameters accordingly. We took care to set $\gamma$ such that our method performs a similar amount of shrinkage compared to other methods, such as IntelPooling, which effectively uses a separate penalty matrix for users and time. To do so, we set a separate value of $\gamma$ for both users and time and set it to the maximum eigenvalue of the penalty matrix (random effect precision matrix) used by IntelPooling. We use 5 neighbors within the DML methods and set the other hyperparameters as follows: $\sigma = 1$, $\lambda = 1$, and $\delta = 0.01$.

For the Neural-Linear method, we generate a $125 \times 125$ array of baseline rewards (no action-specific component) to train the neural network prior to running the bandit algorithm. Consequently, the results shown in the paper for Neural-Linear are better than would be observed in practice because we allowed the Neural-Linear method to leverage data that we did not make available to the other methods. This setup offers the computational benefit of not needing to update the neural network within bandit replications, which substantially reduces the necessary computation time.

Aside from the input features used, the Neural-Linear method has the same implementation as the Standard method. The Neural-Linear method uses the output from the last hidden layer of a neural network to model the baseline reward. However, we use the original features (the state vectors) to model the advantage function because the true advantage function is, in fact, linear in these features.

Our neural networks consisted of four hidden layers with 10, 20, 20, and 10 nodes, respectively. The first two employ the ReLU activation function (Nair & Hinton, 2010) while the latter two employ the hyperbolic tangent. We chose to use the hyperbolic tangent for the last two layers because Snoek et al. (2015) found that smooth activation functions such as the hyperbolic tangent were advantageous in their neural bandit algorithm. The loss function was the mean squared error between the neural networks' output and the baseline reward on a simulated data set. We trained our networks using the Adam optimizer (Kingma & Ba, 2014) with batch sizes of 200 for between 20 and 50 epochs. We simulated a separate validation data set to ensure to check that our model had converged and was generating accurate predictions.

Figure 5 compares the true baseline reward function in the nonlinear setting (left) to that estimated by the neural network (right). We see that neural network produced an accurate approximation of the baseline reward, which helps explain the good performance of the Neural-Linear method relative to other baseline approaches, such as Standard.

We include the Neural-Linear method primarily to demonstrate that correctly modeling the baseline is not sufficient to ensure good performance. In the mobile health context, algorithms should also be able to (1) efficiently pool data across users and time and (2) leverage network information. The Neural-Linear method satisfies neither of these criteria. Note that a neural network could be used to model the baseline rewards as part of our algorithm. Future work could consider allowing the differential rewards themselves to also be complex nonlinear functions, which could be accomplished by combining our method with Neural-Linear. We leave the details to future work.

### D.2 PAIRWISE COMPARISONS FOR MAIN SIMULATION

Table 1 shows pairwise comparisons between methods across the three settings. The individual cells indicate the percentage of repetitions (out of 50) in which the method listed in the row outperformed the method listed in the column. The asterisks indicate p-values below 0.05 from paired two-sided t-tests on the differences in final regret. The Avg column indicates the average pairwise win percentage.

The DML-TS-NNR-BLM and DML-TS-NNR-BT methods perform well across all three settings and the difference between them is statistically indistinguishable. These methods

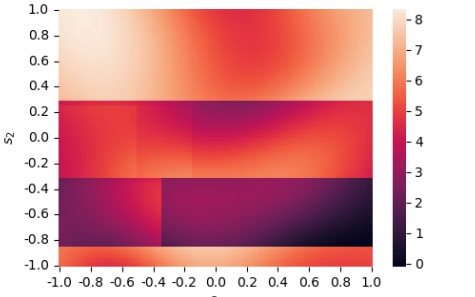 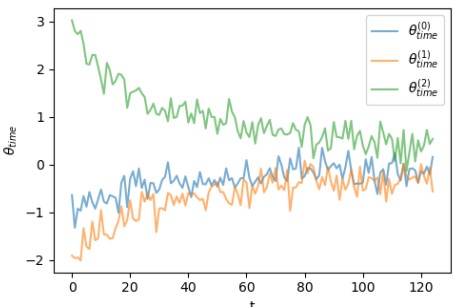

Figure 4: (left) The baseline reward function $g(S_{it})$ used in the simulation study. The proposed method allows this function to be a nonlinear function of the context vectors. The baseline was generated using a combination of recursive partitioning and by summing scaled, shifted, and rotated Gaussian densities. (right) The time-specific parameters used in the simulation study. These parameters cause the advantage function to vary over time. We set them such that the advantage function changes quickly at the beginning of the study then stabilizes.

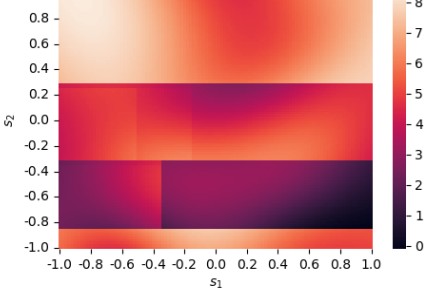 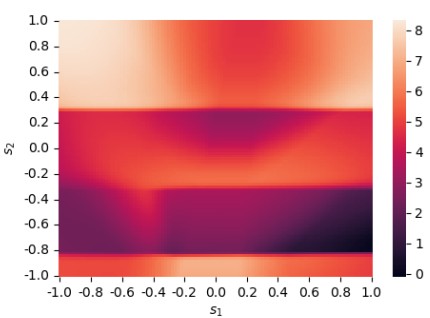

Figure 5: (left) The baseline reward function $g(S_{it})$ used in the simulation study compared to (right) the estimated baseline reward from our neural network in the nonlinear setting.

perform about equally well compared to IntelPooling in the first setting, but they perform better in the other settings because the DML methods (1) can leverage network information and (2) can accurately model the nonlinear baseline reward in the third setting.

DML-TS-NNR-BLM and DML-TS-NNR-BT perform especially well in the nonlinear setting, the more general setting for which they were designed. They achieve pairwise win percentages of 91% and 87%, respectively, compared to only 48% for the next-best non-DML method (IntelPooling).

## Homogeneous Users

|  | 1 | 2 | 3 | 4 | 5 | 6 | 7 | **Avg** |
|---|---|---|---|---|---|---|---|---|
| 1. IntelPooling | - | 54% | 46% | 58% | 100%* | 100%* | 100%* | 76% |
| 2. DML-TS-NNR-BLM | 46% | - | 50% | 52% | 100%* | 100%* | 100%* | 75% |
| 3. DML-TS-NNR-BT | 54% | 50% | - | 58% | 100%* | 100%* | 100%* | 77% |
| 4. DML-TS-SU-BT | 42% | 48% | 42% | - | 100%* | 100%* | 100%* | 72% |
| 5. AC | 0%* | 0%* | 0%* | 0%* | - | 0%* | 0%* | 0% |
| 6. Standard | 0%* | 0%* | 0%* | 0%* | 100%* | - | 0%* | 17% |
| 7. Neural-Linear | 0%* | 0%* | 0%* | 0%* | 100%* | 100%* | - | 33% |

## Heterogeneous Users

|  | 1 | 2 | 3 | 4 | 5 | 6 | 7 | **Avg** |
|---|---|---|---|---|---|---|---|---|
| 1. IntelPooling | - | 0%* | 10%* | 10%* | 100%* | 100%* | 84%* | 51% |
| 2. DML-TS-NNR-BLM | 100%* | - | 54% | 66%* | 100%* | 100%* | 100%* | 87% |
| 3. DML-TS-NNR-BT | 90%* | 46% | - | 64% | 100%* | 100%* | 100%* | 83% |
| 4. DML-TS-SU-BT | 90%* | 34%* | 36% | - | 100%* | 100%* | 100%* | 77% |
| 5. AC | 0%* | 0%* | 0%* | 0%* | - | 0%* | 0%* | 0% |
| 6. Standard | 0%* | 0%* | 0%* | 0%* | 100%* | - | 0%* | 17% |
| 7. Neural-Linear | 16%* | 0%* | 0%* | 0%* | 100%* | 100%* | - | 36% |

## Nonlinear

|  | 1 | 2 | 3 | 4 | 5 | 6 | 7 | **Avg** |
|---|---|---|---|---|---|---|---|---|
| 1. IntelPooling | - | 2%* | 6%* | 24%* | 100%* | 94%* | 62%* | 48% |
| 2. DML-TS-NNR-BLM | 98%* | - | 56% | 94%* | 100%* | 100%* | 100%* | 91% |
| 3. DML-TS-NNR-BT | 94%* | 44% | - | 84%* | 100%* | 100%* | 98%* | 87% |
| 4. DML-TS-SU-BT | 76%* | 6%* | 16%* | - | 100%* | 100%* | 90%* | 65% |
| 5. AC | 0%* | 0%* | 0%* | 0%* | - | 0%* | 0%* | 0% |
| 6. Standard | 6%* | 0%* | 0%* | 0%* | 100%* | - | 10%* | 19% |
| 7. Neural-Linear | 38%* | 0%* | 2%* | 10%* | 100%* | 90%* | - | 40% |

Table 1: Pairwise comparisons between methods in the three settings of the main simulation. Each cell indicates the percent of repetitions (out of 50) in which the method listed in the row outperformed the method listed in the column in term of final regret. Asterisks indicate p-values below 0.05 from paired two-sided t-tests on the differences in final regret. The full DML methods (DML-TS-NNR-BLM and DML-TS-NNR-BT) perform best in all three settings and their final regret is statistically indistinguishable.

### D.3 SIMULATION WITH RECTANGULAR DATA ARRAY

The main simulation involves simulating data from a triangular data array. At the 125-th (final) stage, the algorithm has observed 125 rewards for user 1, 124 rewards for user 2, and so on.

In this section, we simulate actions and rewards under a rectangular array with 100 users and 100 time points. Although we still follow the staged recruitment regime depicted in Figure 3 in Appendix A, at stage 100 we stop sampling actions and rewards for user 1; at stage 101 we stop sampling for user 2; and so on until we have sampled 100 time points for all 100 users. Aside from the shape of the data array, the setup is the same for this simulation as for the main simulation.

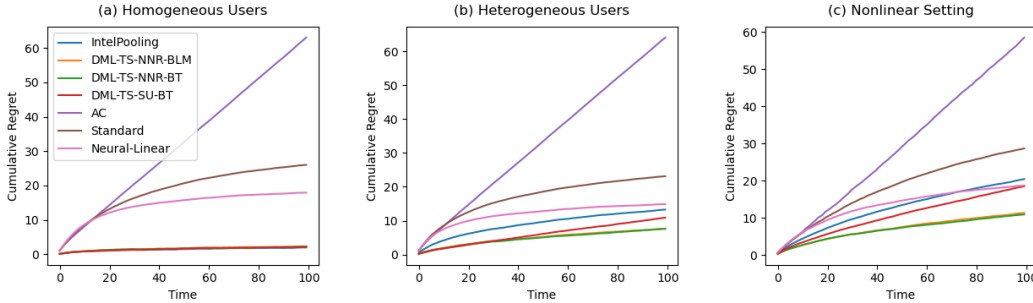

Figure 6: Cumulative regret in the (a) Homogeneous Users, (b) Heterogeneous Users, and (c) Nonlinear settings using a rectangular array of data in which we observe 100 time points for 100 users in a stagewise fashion as depicted in Figure 3 in Appendix A. Similar to Figure 1, the full DML methods (DML-TS-NNR-BLM and DML-TS-NNR-BT) are highly competitive in all three settings and substantially outperform the other methods in the nonlinear setting.

The cumulative regret for these methods as a function of time—not stage—is shown in Figure 6. The qualitative results are quite similar to those from the main simulation. DML-TS-NNR-BLM and DML-TS-NNR-BT are highly competitive in all three scenarios and substantially outperform the other methods in the nonlinear setting; in fact, compared to the main simulation, the improvement of these methods over the others is even larger.

Table 2 displays the results from pairwise method comparisons, similar to those shown in Table 1. Again the results are qualitatively similar to those from the main simulation. In 26/30 pairwise comparisons, DML-TS-NNR-BLM and DML-TS-NNR-BT outperform the other methods in 100% of replications. The four remaining pairwise comparisons are not statistically significant and result from comparisons with DML-TS-SU-BT and IntelPooling in the "Homogeneous Users" setting. DML-TS-NNR-BLM and DML-TS-NNR-BT offer little or no benefit compared to these methods in that setting because (1) the baseline is linear, (2) no network information is available, (3) no time effects are present, and (4) partial pooling (employed by DML-TS-NNR-BLM, DML-TS-NNR-BT, and IntelPooling) offers no benefit compared to full pooling (employed by DML-TS-SU-BT) because the causal effects are exactly the same among users.

In summary, our methods substantially outperform the other methods in complex, general settings and perform competitively with the other methods in simple settings.

## E  Additional Details for Valentine Study

Personalizing treatment delivery in mobile health is a common application for online learning algorithms. We focus here on the Valentine study, a prospective, randomized-controlled, remotely administered trial designed to evaluate an mHealth intervention to supplement cardiac rehabilitation for low- and moderate-risk patients (Jeganathan et al., 2022). We aim to use smart watch data (Apple Watch and Fitbit) obtained from the Valentine study to learn the optimal timing of notification delivery given the users' current context.

### E.1  Data from the Valentine Study

Prior to the start of the trial, baseline data was collected on each of the participants (e.g., age, gender, baseline activity level, and health information). During the study, participants

## Homogeneous Users

|  | 1 | 2 | 3 | 4 | 5 | 6 | 7 | **Avg** |
|---|---|---|---|---|---|---|---|---|
| 1. IntelPooling | - | 62% | 60% | 58% | 100%* | 100%* | 100%* | 80% |
| 2. DML-TS-NNR-BLM | 38% | - | 50% | 44% | 100%* | 100%* | 100%* | 72% |
| 3. DML-TS-NNR-BT | 40% | 50% | - | 44% | 100%* | 100%* | 100%* | 72% |
| 4. DML-TS-SU-BT | 42% | 56% | 56% | - | 100%* | 100%* | 100%* | 76% |
| 5. AC | 0%* | 0%* | 0%* | 0%* | - | 0%* | 0%* | 0% |
| 6. Standard | 0%* | 0%* | 0%* | 0%* | 100%* | - | 0%* | 17% |
| 7. Neural-Linear | 0%* | 0%* | 0%* | 0%* | 100%* | 100%* | - | 33% |

## Heterogeneous Users

|  | 1 | 2 | 3 | 4 | 5 | 6 | 7 | **Avg** |
|---|---|---|---|---|---|---|---|---|
| 1. IntelPooling | - | 0%* | 0%* | 8%* | 100%* | 100%* | 86%* | 49% |
| 2. DML-TS-NNR-BLM | 100%* | - | 48% | 100%* | 100%* | 100%* | 100%* | 91% |
| 3. DML-TS-NNR-BT | 100%* | 52% | - | 100%* | 100%* | 100%* | 100%* | 92% |
| 4. DML-TS-SU-BT | 92%* | 0%* | 0%* | - | 100%* | 100%* | 100%* | 65% |
| 5. AC | 0%* | 0%* | 0%* | 0%* | - | 0%* | 0%* | 0% |
| 6. Standard | 0%* | 0%* | 0%* | 0%* | 100%* | - | 0%* | 17% |
| 7. Neural-Linear | 14%* | 0%* | 0%* | 0%* | 100%* | 100%* | - | 36% |

## Nonlinear

|  | 1 | 2 | 3 | 4 | 5 | 6 | 7 | **Avg** |
|---|---|---|---|---|---|---|---|---|
| 1. IntelPooling | - | 0%* | 0%* | 12%* | 100%* | 100%* | 18%* | 38% |
| 2. DML-TS-NNR-BLM | 100%* | - | 44% | 100%* | 100%* | 100%* | 100%* | 91% |
| 3. DML-TS-NNR-BT | 100%* | 56% | - | 100%* | 100%* | 100%* | 100%* | 93% |
| 4. DML-TS-SU-BT | 88%* | 0%* | 0%* | - | 100%* | 100%* | 64% | 59% |
| 5. AC | 0%* | 0%* | 0%* | 0%* | - | 0%* | 0%* | 0% |
| 6. Standard | 0%* | 0%* | 0%* | 0%* | 100%* | - | 0%* | 17% |
| 7. Neural-Linear | 82%* | 0%* | 0%* | 36% | 100%* | 100%* | - | 53% |

Table 2: Pairwise comparisons between methods in the three settings of the simulation with a rectangular array of data. As in Table 1, each cell indicates the percent of repetitions (out of 50) in which the method listed in the row outperformed the method listed in the column in term of final regret. Asterisks indicate p-values below 0.05 from paired two-sided t-tests on the differences in final regret. The full DML methods (DML-TS-NNR-BLM and DML-TS-NNR-BT) perform best in all three settings in terms of pairwise win percentages and are not significantly different from each other.

are randomized to either receive a notification ($A_t = 1$) or not ($A_t = 0$) at each of 4 daily time points (morning, lunchtime, mid-afternoon, evening), with probability 0.25. Contextual information was collected frequently (e.g., number of messages sent in prior week, step count variability in prior week, and pre-decision point step-counts).

Since the goal of the Valentine study is to increase participants' activity levels, we thereby define the reward, $R_t$, as the step count for the 60 minutes following a decision point (log-transformed to eliminate skew). Our application also uses a subset of the baseline and contextual data; this subset contains the variables with the strongest association to the reward. Table 3 shows the features available to the bandit in the Valentine study data set.

| Feature | Description | Interaction | Baseline Model |
|---|---|---|---|
| Phase II | 1 if in Phase II, 0 o.w. | $\checkmark$ | $\checkmark$ |
| Phase III | 1 if in Phase II, 0 o.w. | $\checkmark$ | $\checkmark$ |
| Steps in prior 30 minutes | log transformed | $\checkmark$ | $\checkmark$ |
| Pre-trial average daily steps | log transformed | $\times$ | $\checkmark$ |
| Device | 1 if Fitbit, 0 o.w. | $\times$ | $\checkmark$ |
| Prior week step count variability | SD of the rewards in prior week | $\times$ | $\checkmark$ |

Table 3: List of features available to the bandit in the Valentine study. The features available to model the action interaction (effect of sending an anti-sedentary message) and to model the baseline (reward under no action) are denoted via a "$\checkmark$" in the corresponding column, otherwise $\times$.

For baseline variables, we use the participant's device model ($Z_1$, Fitbit coded as 1), the participant's step count variability in the prior week ($Z_2$), and a measure of the participant's pre-trial activity level based on an intake survey ($Z_3$, with larger values corresponding to higher activity levels).

At every decision point, before selecting an action, the learner sees two state variables: the participant's previous 30-minute step count ($S_1$, log-transformed) and the participant's phase of cardiac rehabilitation ($S_2$, dummy coded). The cardiac rehabilitation phase is defined based on a participant's time in the study: month 1 represents Phase I, month 2-4 represents Phase II, and month 5-6 represents Phase III.

### E.2    EVALUATION

The Valentine study collected the sensor-based features at 4 decision points per day for each study participant. The reward for each message was defined to be $\log(0.5 + x)$, where $x$ is the step count of the participant in the 60 minutes following the notification. As noted in the introduction, the baseline reward, i.e. the step count of a subject when no message is sent, not only depends on the state in a complex way but is likely dependent on a large number of time-varying observed variables. Both these characteristics (complex, time-varying baseline reward function) suggest using our proposed approach.

We generated 100 bootstrap samples and ran our contextual bandit on them, considering the binary action of whether or not to send a message at a given decision point based on the contextual variables $S_1$ and $S_2$. Each user is considered independently and with a cohesion network, for maximum personalization and independence of results. To guarantee that messages have a positive probability of being sent, we only sample the observations with notification randomization probability between 0.01 and 0.99. In the case of the algorithm utilizing NNR, we chose four baseline characteristics (gender, age, device, and baseline average daily steps) to establish a measure of "distance" between users. For this analysis, the value of $k$ representing the number of nearest neighbors was set to 5. To utilize bootstrap sampling, we train the Neural-Linear method's neural network using out-of-bag samples. The neural network architecture comprises a single hidden layer with two hidden nodes. The input contains both the baseline characteristics and the contextual variables and the activation function applied here is the *softplus* function, defined as $\text{softplus}(x) = \log(1 + \exp(x))$.

We performed an offline evaluation of the contextual bandit algorithms using an inverse propensity score (IPS) version of the method from Li et al. (2010), where the sequence of states, actions, and rewards in the data are used to form a near-unbiased estimate of the average expected reward achieved by each algorithm, averaging over all users.

### E.3    Inverse Propensity Score (IPS) offline evaluation

In the implemented Valentine study, the treatment was randomized with a constant probability $p_t = 0.25$ at each time $t$. To conduct off-policy evaluation using our proposed algorithm and the competing variations of the TS algorithm, we outline the IPS estimator for an unbiased estimate of the per-trial expected reward based on what has been studied in Li et al. (2010).

Given the logged data $\mathcal{D} = \{s_t = s_t, A_t = a_t, R_t = r_t\}_{t=1}^T$ collected under the policy $\mathbf{p} = \{p_t\}_{t=1}^T$, and the treatment policy being evaluated $\pi = \{\pi_t\}_{t=1}^T$, the objective of this offline estimator is to reweight the observed reward sequence $\{R_t\}_{t=1}^T$ to assign varying importance to actions based on the propensities of both the original and new policies in selecting them.

**Lemma 1** (Unbiasedness of the IPS estimator). *Assuming the positivity assumption in logging, which states that for any given $s$ and $a$, if $p_t(a|s) > 0$, then we also have $\pi_t(a|s) > 0$, we can obtain an unbiased per-trial expected reward using the following IPS estimator:*

$$\hat{R}_{IPS} = \frac{1}{T} \sum_{t=1}^T \frac{\pi_t(a_t|s_t)}{p_t(a_t|s_t)} r_t \tag{6}$$

As mentioned in the previous section, we restrict our sampling to observations with notification randomization probabilities ranging from 0.01 to 0.99. This selection criterion ensures the satisfaction of the positivity assumption. The proof essentially follows from definition, we have:

*Proof.*

$$\mathbb{E}[R_{\text{IPS}}] = \mathbb{E}_{\mathbf{p}} \left[ \frac{1}{T} \sum_{t=1}^T \frac{\pi_t(a_t|s_t)}{p_t(a_t|s_t)} R_t(a_t, s_t) \right]$$

$$= \frac{1}{T} \sum_{t=1}^T \frac{\pi_t(a_t|s_t)}{p_t(a_t|s_t)} R_t(a_t, s_t) \times p_t(a_t|s_t)$$

$$= \frac{1}{T} \sum_{t=1}^T \pi_t(a_t|s_t) R_t(a_t, s_t)$$

$$= \mathbb{E}_\pi \left[ \frac{1}{T} \sum_{t=1}^T R_t(a_t, s_t) \right]$$

$\square$

To address the instability issue caused by re-weighting in some cases, we use a Self-Normalized Inverse Propensity Score (SNIPS) estimator. This estimator scales the results by the empirical mean of the importance weights, and still maintains the property of unbiasedness.

$$\hat{R}_{\text{SNIPS}} = \frac{\hat{R}_{\text{IPS}}}{\frac{1}{T} \sum_{t=1}^T \frac{\pi_t(a_t|s_t)}{p_t(a_t|s_t)}} = \frac{\sum_{t=1}^T \frac{\pi_t(a_t|s_t)}{p_t(a_t|s_t)} r_t}{\sum_{t=1}^T \frac{\pi_t(a_t|s_t)}{p_t(a_t|s_t)}} \tag{7}$$

## F    Additional Details for the Intern Health Study (IHS)

To further enhance the competitive performance of our proposed DML-TS-NNR algorithm, we conducted an additional comparative analysis using a real-world dataset from the Intern

Health Study (IHS) (NeCamp et al., 2020). This micro-randomized trial investigated the use of mHealth interventions aimed at improving the behavior and mental health of individuals in stressful work environments. The estimates obtained represent the improvement in average reward relative to the original constant randomization, averaging across stages (K = 30) and participants (N = 1553). The available IHS data consist of 20 multiple-imputed data sets. We apply the algorithms to each imputed data set and perform a comparative analysis of the competing algorithms. The results presented in Figure 7 shows our proposed DML-TS-NNR-RF algorithm achieved significantly higher rewards than the other three competing ones and demonstrated comparable performance to the AC algorithm. These findings further support the advantages of our proposed algorithm.

## F.1 Data from the IHS

Prior to the start of the trial, baseline data was collected on each of the participants (e.g., institution, specialty, gender, baseline activity level, and health information). During the study, participants are randomized to either receive a notification ($A_t = 1$) or not ($A_t = 0$) every day, with probability 3/8. Contextual information was collected frequently (e.g., step count in prior five days, and current day in study).

We hereby define the reward, $R_t$, as the step count on the following day (cubic root). Our application also uses a subset of the baseline and contextual data; this subset contains the variables with the strongest association to the reward. Table 4 shows the features available to the bandit in the IHS data set.

| Feature | Description | Interaction | Baseline |
|---|---|---|---|
| Day in study | an integer from 1 to 30 | $\checkmark$ | $\checkmark$ |
| Average daily steps in prior five days | cubic root | $\checkmark$ | $\checkmark$ |
| Average daily sleep in prior five days | cubic root | $\times$ | $\checkmark$ |
| Average daily mood in prior five days | a Likert scale from $1-10$ | $\times$ | $\checkmark$ |
| Pre-intern average daily steps | cubic root | $\times$ | $\checkmark$ |
| Pre-intern average daily sleep | cubic root | $\times$ | $\checkmark$ |
| Pre-intern average daily mood | a Likert scale from $1-10$ | $\times$ | $\checkmark$ |
| Sex | Gender | $\times$ | $\checkmark$ |
| Week category | The theme of messages in a specific week (mood, sleep, activity, or none) | $\times$ | $\checkmark$ |
| PHQ score | PHQ total score | $\times$ | $\checkmark$ |
| Early family environment | higher score indicates higher level of adverse experience | $\times$ | $\checkmark$ |
| Personal history of depression | | $\times$ | $\checkmark$ |
| Neuroticism (Emotional experience) | higher score indicates higher level of neuroticism | $\times$ | $\checkmark$ |

Table 4: List of features available to the bandit in the IHS. The features available to model the action interaction (effect of sending a mobile prompt) and to model the baseline (reward under no action) are denoted via a "$\checkmark$" in the corresponding column, otherwise $\times$.

At every decision point, before selecting an action, the learner sees two state variables: the participant's previous 5-day average daily step count ($S_1$, cubic root) and the participant's day in study ($S_2$, an integer from 1 to 30).

## F.2 Evaluation

We run our contextual bandit on the IHS data, considering the binary action of whether or not to send a message at a given decision point based on the contextual variables $S_1$

and $S_2$. Each user is considered independently and with a cohesion network, for maximum personalization and independence of results. To guarantee that messages have a positive probability of being sent, we only sample the observations with notification randomization probability between 0.01 and 0.99. For the algorithm employing NNR, we defined participants in the same institution as their own "neighbors". This definition enables the flexibility for the value of $k$, representing the number of nearest neighbors, to vary for each participant based on their specific institutional context. Furthermore, in our study setting, we make the assumption that individuals from the same "institution" enter the study simultaneously as a group. Due to the limited access to prior data, we are unable to build the neural linear models as in the Valentine Study.

We utilized 20 multiple-imputed data sets and performed an offline evaluation of the contextual bandit algorithms on each data set. The result is presented below in Figure 7.

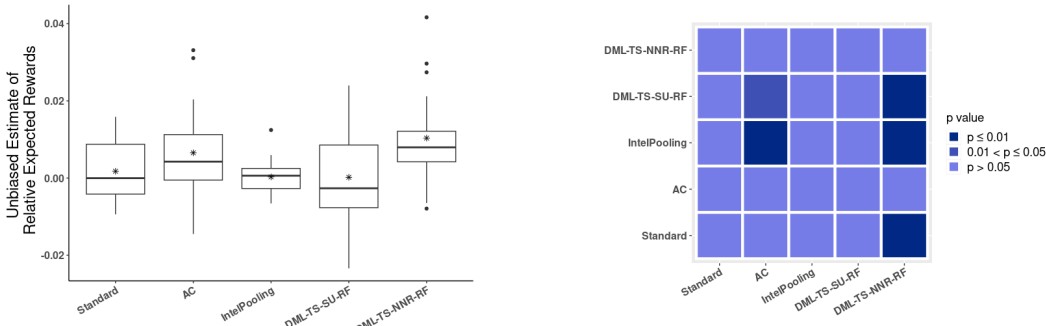

Figure 7: **(left)** Unbiased estimates of the average per-trial reward for all five competing algorithms, relative to the reward obtained under the pre-specified Valentine randomization policy across 20 multiple-imputed data sets. And **(right)** p-values from the pairwise paired t-tests. The dark shade in the last column indicates that the proposed DML-TS-NNR-RF algorithm achieved significantly higher rewards than the other three competing algorithms while demonstrating comparable performance to the AC algorithm.

## G  REGRET BOUND

### G.1  DOUBLE ROBUSTNESS OF PSEUDO-REWARD

**Lemma 2.** *If either $p_{i,t} = \pi_{i,t}$ or $f_{i,t} = r_{i,t}$, then*

$$\mathbb{E}\left[\tilde{R}_{i,t}^f | s, \bar{a}\right] = \Delta_{i,t}(s, \bar{a}).$$

*That is, the pseudo-reward is an unbiased estimator of the true differential reward.*

*Proof.* Recall that

$$\tilde{R}_{i,t}^f = \frac{R_{it} - f_{i,t}(s, A_{i,t})}{\delta_{A_{i,t}=\bar{a}} - \pi_{i,t}(0|s)} + \Delta_{i,t}^f(s, \bar{a})$$

**Case I: $\pi$'s are correctly specified**

Then

$$\mathbb{E}\left[\frac{R_{it}}{\delta_{A_{i,t}=\bar{a}} - \pi_{i,t}(0|s)}\bigg| s, \bar{a}\right] = r_{i,t}(s, \bar{a}) - r_{i,t}(s, 0)$$

$$= \Delta_{i,t}(s, \bar{a})$$

and

$$\mathbb{E}\left[\frac{f_{i,t}(s, A_{i,t})}{\delta_{A_{i,t}=\bar{a}} - \pi_{i,t}(0|s)}\Bigg| s, \bar{a}\right] = f_{i,t}(s, \bar{a}) - f_{i,t}(s, 0)$$

$$= \Delta_{i,t}^f(s, \bar{a})$$

so that

$$\mathbb{E}\left[\frac{R_{it} - f_{i,t}(s, A_{i,t})}{\delta_{A_{i,t}=\bar{a}} - \pi_{i,t}(0|s)} + \Delta_{i,t}^f(s, \bar{a})\Bigg| s, \bar{a}\right] = \Delta_{i,t}(s, \bar{a}) - \Delta_{i,t}^f(s, \bar{a}) + \Delta_{i,t}^f(s, \bar{a})$$

$$= \Delta_{i,t}(s, \bar{a})$$

**Case II: $f$ correctly specified**

$$\mathbb{E}\left[\frac{R_{it}}{\delta_{A_{i,t}=\bar{a}} - \pi_{i,t}(0|s)}\Bigg| s, \bar{a}\right] = \frac{1 - p_{i,t}(0|s)}{1 - \pi_{i,t}(0|s)}r_{i,t}(s, \bar{a}) - \frac{p_{i,t}(0|s)}{\pi_{i,t}(0|s)}r_{i,t}(s, 0)$$

and

$$\mathbb{E}\left[\frac{f_{i,t}(s, A_{i,t})}{\delta_{A_{i,t}=\bar{a}} - \pi_{i,t}(0|s)}\Bigg| s, \bar{a}\right] = \frac{1 - p_{i,t}(0|s)}{1 - \pi_{i,t}(0|s)}f_{i,t}(s, \bar{a}) - \frac{p_{i,t}(0|s)}{\pi_{i,t}(0|s)}f_{i,t}(s, \bar{0})$$

$$= \frac{1 - p_{i,t}(0|s)}{1 - \pi_{i,t}(0|s)}r_{i,t}(s, \bar{a}) - \frac{p_{i,t}(0|s)}{\pi_{i,t}(0|s)}r_{i,t}(s, \bar{0})$$

and

$$\mathbb{E}\left[\Delta_{i,t}^f(s, \bar{a})\Big| s, \bar{a}\right] = \Delta_{i,t}(s, \bar{a})$$

$$\mathbb{E}\left[\frac{R_{it} - f_{i,t}(s, \bar{A}_{i,t})}{\delta_{A_{i,t}=\bar{a}} - \pi_{i,t}(0|s)} + \Delta_{i,t}^f(s, \bar{a})\Bigg| s, \bar{a}\right] = \Delta_{i,t}(s, \bar{a})$$

$\square$

## G.2 PRELIMINARIES

**Lemma 3.** *Let $X$ be a mean-zero sub-Gaussian random variable with variance factor $v^2$ and $Y$ be a bounded random variable such that $|Y| \leq B$ for some $0 \leq B < \infty$. Then $XY$ is sub-Gaussian with variance factor $v^2 B^2$.*

*Proof.* Recall that $X$ being mean-zero sub-Gaussian means that

$$P(|X| \geq t) \leq 2\exp\left(-\frac{t^2}{2v^2}\right).$$

Now note that

$$|XY| \leq |X|B$$

so that if $|XY| > t$, then $|X|B > t$. Thus by monotonicity

$$P(|XY| \geq t) \leq P(|X|B \geq t)$$

$$= P\left(|X| \geq \frac{t}{B}\right)$$

$$\leq 2\exp\left(-\frac{t^2}{2B^2 v^2}\right)$$

as desired. $\square$

**Lemma 4.** *If $X, Y$ are sub-Gaussian with variance factors $v_x^2$, $v_y^2$, respectively, then $\alpha X + \beta Y$ is sub-Gaussian with variance factor $\alpha^2 v_x^2 + \beta^2 v_y^2 \forall \alpha, \beta \in \mathbb{R}$.*

*Proof.* Recall the equivalent definition of sub-Gaussianity that $X, Y$ are sub-Gaussian iff for some $b, a > 0$ and all $\lambda > 0$

$$\mathbb{E} \exp\left(\lambda(X - \mathbb{E}X)\right) \leq \exp(\lambda^2 v_x^2 / 2)$$
$$\mathbb{E} \exp\left(\lambda(Y - \mathbb{E}Y)\right) \leq \exp(\lambda^2 v_y^2 / 2)$$

Then

$$\mathbb{E} \exp\left(\lambda(\alpha X + \beta Y - \alpha \mathbb{E}X - \beta \mathbb{E}Y)\right) \leq \sqrt{\mathbb{E} \exp\left(2\alpha\lambda(X - \mathbb{E}X)\right)} \sqrt{\mathbb{E} \exp\left(2\beta\lambda(Y - \mathbb{E}Y)\right)}$$
$$\leq \sqrt{\exp(2\alpha^2\lambda^2 v_x^2)} \sqrt{\exp(2\beta^2\lambda^2 v_y^2)}$$
$$= \exp((\alpha^2 v_x^2 + \beta^2 v_y^2)\lambda^2)$$

$\square$

The following Lemma gives the sub-Gaussianity and variance of the difference between the pseudo-reward and its expectation. We see that in the variance, all terms except those involving the inverse propensity weighted noise variance vanish as $f_{i,t}$ becomes a better estimate of $r_{i,t}$, as long as $f_{i,t}$ is uniformly bounded. Note that means and variances may be implicitly conditioned on the history.

**Lemma 5.** *If $\pi_{i,t}$ is correctly specified and $\tilde{\sigma}_{i,t}^2 \geq \frac{1}{c}$, the difference between the pseudo-reward and its expectation (taken wrt the action and noise) is mean zero sub-Gaussian with variance*

$$v^2 \equiv \frac{(r_{i,t}(s,\bar{a}) - f_{i,t}(s,\bar{a}))^2 + Var(\epsilon_{i,t})}{1 - \pi_{i,t}(0|s)} + \frac{(r_{i,t}(s,0) - f_{i,t}(s,0))^2 + Var(\epsilon_{i,t})}{\pi_{i,t}(0|s)}$$
$$+ 2(\Delta_{i,t}(s,\bar{a}) - \Delta_{i,t}^f(s,\bar{a}))\Delta_{i,t}^f(s,\bar{a})$$

*Proof.* We need to show that it is sub-Gaussian and upper bound its variance. We write the difference as

$$\tilde{R}_{i,t}^f(s,\bar{a}) - \mathbb{E}[\tilde{R}_{i,t}^f|s,\bar{a}] = \tilde{R}_{i,t}^f(s,\bar{a}) - \Delta_{i,t}(s,\bar{a})$$
$$= \frac{R_{it} - f_{i,t}(s,A_{i,t})}{\delta_{A_{i,t}=\bar{a}} - \pi_{i,t}(0|s)} + \Delta_{i,t}^f(s,\bar{a}) - \Delta_{i,t}(s,\bar{a})$$
$$= \frac{r_{i,t}(s,A_{i,t}) - f_{i,t}(s,A_{i,t}) + \epsilon_{i,t}}{\delta_{A_{i,t}=\bar{a}} - \pi_{i,t}(0|s)} + \Delta_{i,t}^f(s,\bar{a}) - \Delta_{i,t}(s,\bar{a})$$

Note that $|r_{i,t}(s,A_{i,t})| \leq \max\left(|r_{i,t}(s,\bar{a})|, |r_{i,t}(s,0)|\right)$ and $|f_{i,t}(s,A_{i,t})| \leq \max\left(|f_{i,t}(s,\bar{a})|, |f_{i,t}(s,0)|\right)$. Thus since $\left|\frac{1}{\delta_{A_{i,t}=\bar{a}} - \pi_{i,t}(0|s)}\right|$ is upper bounded by $c > 0$, we have that $\frac{r_{i,t}(s,A_{i,t}) - f_{i,t}(s,A_{i,t})}{\delta_{A_{i,t}=\bar{a}} - \pi_{i,t}(0|s)}$ is bounded and thus (not necessarily mean zero) sub-Gaussian. Since $\epsilon_{i,t}$ is sub-Gaussian, its denominator is bounded, and the remaining terms are deterministic, the entire difference between the pseudo-reward and its mean is sub-Gaussian. Now

$$\text{Var}(\tilde{R}_{i,t}^f(s,\bar{a}) - \mathbb{E}[\tilde{R}_{i,t}^f|s,\bar{a}]) = \text{Var}\left(\tilde{R}_{i,t}^f(s,\bar{a})\right)$$
$$= \mathbb{E}\left[\tilde{R}_{i,t}^f(s,\bar{a})^2\right] - \Delta_{i,t}(s,\bar{a})^2 \tag{8}$$

since $\mathbb{E}[\tilde{R}_{i,t}^f|s,\bar{a}]$ is not random. Now we expand the first term on the rhs.

$$
\begin{aligned}
\mathbb{E}\left[\tilde{R}_{i,t}^f(s,\bar{a})^2\right] &= \mathbb{E}\left[\left(\frac{r_{i,t}(s,A_{i,t}) - f_{i,t}(s,A_{i,t}) + \epsilon_{i,t}}{\delta_{A_{i,t}=\bar{a}} - \pi_{i,t}(0|s)} + \Delta_{i,t}^f(s,\bar{a})\right)^2\right] \\
&= \mathbb{E}\left[\left(\frac{r_{i,t}(s,A_{i,t}) - f_{i,t}(s,A_{i,t}) + \epsilon_{i,t}}{\delta_{A_{i,t}=\bar{a}} - \pi_{i,t}(0|s)}\right)^2\right] \\
&\quad + 2\mathbb{E}\left[\frac{r_{i,t}(s,A_{i,t}) - f_{i,t}(s,A_{i,t}) + \epsilon_{i,t}}{\delta_{A_{i,t}=\bar{a}} - \pi_{i,t}(0|s)}\right]\Delta_{i,t}^f(s,\bar{a}) + \Delta_{i,t}^f(s,\bar{a})^2 \\
&= \mathbb{E}\left[\left(\frac{r_{i,t}(s,A_{i,t}) - f_{i,t}(s,A_{i,t}) + \epsilon_{i,t}}{\delta_{A_{i,t}=\bar{a}} - \pi_{i,t}(0|s)}\right)^2\right] \\
&\quad + 2(\Delta_{i,t}(s,\bar{a}) - \Delta_{i,t}^f(s,\bar{a}))\Delta_{i,t}^f(s,\bar{a}) + \Delta_{i,t}^f(s,\bar{a})^2 \quad (9)
\end{aligned}
$$

For the first term on the rhs of Eqn. 9,

$$
\begin{aligned}
&\mathbb{E}\left[\left(\frac{r_{i,t}(s,A_{i,t}) - f_{i,t}(s,A_{i,t}) + \epsilon_{i,t}}{\delta_{A_{i,t}=\bar{a}} - \pi_{i,t}(0|s)}\right)^2\right] \\
&= \mathbb{E}\left[\left(\frac{r_{i,t}(s,A_{i,t}) - f_{i,t}(s,A_{i,t})}{\delta_{A_{i,t}=\bar{a}} - \pi_{i,t}(0|s)}\right)^2\right] + \mathbb{E}\left[\left(\frac{\epsilon_{i,t}}{\delta_{A_{i,t}=\bar{a}} - \pi_{i,t}(0|s)}\right)^2\right] \\
&= \frac{(r_{i,t}(s,\bar{a}) - f_{i,t}(s,\bar{a}))^2 + \mathbb{E}[\epsilon_{i,t}^2]}{1 - \pi_{i,t}(0|s)} + \frac{(r_{i,t}(s,0) - f_{i,t}(s,0))^2 + \mathbb{E}[\epsilon_{i,t}^2]}{\pi_{i,t}(0|s)}
\end{aligned}
$$

so that plugging this into Eqn. 9, we have

$$
\begin{aligned}
\mathbb{E}\left[\tilde{R}_{i,t}^f(s,\bar{a})^2\right] &= \frac{(r_{i,t}(s,\bar{a}) - f_{i,t}(s,\bar{a}))^2 + \mathbb{E}[\epsilon_{i,t}^2]}{1 - \pi_{i,t}(0|s)} + \frac{(r_{i,t}(s,0) - f_{i,t}(s,0))^2 + \mathbb{E}[\epsilon_{i,t}^2]}{\pi_{i,t}(0|s)} \\
&\quad + 2(\Delta_{i,t}(s,\bar{a}) - \Delta_{i,t}^f(s,\bar{a}))\Delta_{i,t}^f(s,\bar{a}) + \Delta_{i,t}^f(s,\bar{a})^2
\end{aligned}
$$

and plugging this into Eqn. 8 we obtain the variance.

$$
\begin{aligned}
\mathrm{Var}\left(\tilde{R}_{i,t}^f(s,\bar{a})\right) &= \frac{(r_{i,t}(s,\bar{a}) - f_{i,t}(s,\bar{a}))^2 + \mathbb{E}[\epsilon_{i,t}^2]}{1 - \pi_{i,t}(0|s)} + \frac{(r_{i,t}(s,0) - f_{i,t}(s,0))^2 + \mathbb{E}[\epsilon_{i,t}^2]}{\pi_{i,t}(0|s)} \\
&\quad + 2(\Delta_{i,t}(s,\bar{a}) - \Delta_{i,t}^f(s,\bar{a}))\Delta_{i,t}^f(s,\bar{a}) \\
&= \frac{(r_{i,t}(s,\bar{a}) - f_{i,t}(s,\bar{a}))^2 + \mathrm{Var}(\epsilon_{i,t})}{1 - \pi_{i,t}(0|s)} + \frac{(r_{i,t}(s,0) - f_{i,t}(s,0))^2 + \mathrm{Var}(\epsilon_{i,t})}{\pi_{i,t}(0|s)} \\
&\quad + 2(\Delta_{i,t}(s,\bar{a}) - \Delta_{i,t}^f(s,\bar{a}))\Delta_{i,t}^f(s,\bar{a})
\end{aligned}
$$

as desired. $\qquad\square$

The next corollary follows immediately and shows the variance factor's stochastic convergence rate to the scaled variance factor of the noise.

**Corollary 1.** *If $\|f_{i,t} - r_{i,t}\|_\infty = \tilde{O}_P(k^{-1/4})$, where $\|\cdot\|_\infty$ is the $L^\infty$ norm, and $f_{i,t}$ is uniformly bounded, then*

$$
v_k^2 = c\tilde{O}_P(k^{-1/2}) + \sigma^2 c^2
$$

In the next remark, we show what the variance would be if we did *not* use DML and estimate $f_{i,t} \approx r_{i,t}$, but used only the inverse propensity weighted observed reward as the pseudo-reward. This was done in Greenewald et al. (2017). In this case, there are terms dependent on the mean reward that do *not* vanish as the number of stages goes to infinity.

**Remark 2.** *If we instead used as our pseudo-reward the inverse propensity weighted observed reward*

$$
\tilde{R}_{i,t}(s,\bar{a}) = \frac{R_{it}}{\delta_{A_{i,t}=\bar{a}} - \pi_{i,t}(0|s)}
$$

*this would be unbiased with variance*

$$v_k^2 \equiv \frac{r_{i,t}(s,\bar{a})^2 + Var(\epsilon_{i,t})}{1 - \pi_{i,t}(0|s)} + \frac{r_{i,t}(s,0)^2 + Var(\epsilon_{i,t})}{\pi_{i,t}(0|s)}$$

*Proof.* The unbiasedness is clear from our proof of Lemma 2. For the variance,

$$\text{Var}\left(\frac{R_{it}}{\delta_{A_{i,t}=\bar{a}} - \pi_{i,t}(0|s)}\right) = \mathbb{E}\left[\frac{(r_{i,t}(s,\bar{a}) + \epsilon_{i,t})^2}{1 - \pi_{i,t}(0|s)} + \frac{(r_{i,t}(s,0) + \epsilon_{i,t})^2}{\pi_{i,t}(0|s)}\right]$$

$$= \frac{r_{i,t}(s,\bar{a})^2 + \text{Var}(\epsilon_{i,t})}{1 - \pi_{i,t}(0|s)} + \frac{r_{i,t}(s,0)^2 + \text{Var}(\epsilon_{i,t})}{\pi_{i,t}(0|s)}$$

$\square$

Here we collect three important results. For analysis at stage $K$, let $\phi(x_{i,t})$ encodes the vector $x_{i,t}$ for the $i$th individual and the $t$th time in a vector of length $2d*K$. At stage $k$, let $\mathcal{O}_k$ denote the set of observed time points across all individuals at stage $k$. Let $\hat{\theta}_k$ denote the estimates for stage $k+1$ using data from $\mathcal{O}_k$. First, we prove a slightly modified version of Lemma 10 from Abbasi-Yadkori et al. (2011):

**Lemma 6.** *(Determinant-Trace Inequality) Suppose $\tilde{\sigma}_{i,t}\phi(\boldsymbol{x}_{i,t}) \in \mathbb{R}^d$. Let $V_{k+1} = \sum_{(i,t)\in\mathcal{O}_k} \tilde{\sigma}_{i,t}^2 \phi(\boldsymbol{x}_{i,t})\phi(\boldsymbol{x}_{i,t})^\top + V_0$. Then*

$$\det(V_{k+1}) \leq \left(\frac{4tr(V_0) + |\mathcal{O}_k|}{4d}\right)^d$$

*Proof.* Following the arguments for the proof of the original Lemma 11 in Abbasi-Yadkori et al. (2011), we have

$$\det(V_{k+1}) \leq \left(\frac{\text{tr}(V_{k+1})}{d}\right)^d.$$

Now we have $\text{tr}(\tilde{\sigma}_{i,t}^2\phi(\boldsymbol{x}_{i,t})\phi(\boldsymbol{x}_{i,t})^\top) \leq 1/4$ since $\|x_{i,t}\| \leq 1$ and $\tilde{\sigma}_{i,t}^2 \leq (1/2)^2$.

$$\text{tr}(V_{k+1}) = \text{tr}(V_0) + \sum_{(i,t)\in\mathcal{O}_k} \tilde{\sigma}_{i,t}^2 \text{tr}(\phi(\boldsymbol{x}_{i,t})\phi(\boldsymbol{x}_{i,t})^\top)$$

$$= \text{tr}(V_0) + \sum_{(i,t)\in\mathcal{O}_k} \tilde{\sigma}_{i,t}^2 \|\phi(\boldsymbol{x}_{i,t})\|_2^2$$

$$\leq \text{tr}(V_0) + \frac{1}{4}|\mathcal{O}_k|$$

Plugging this in to the rhs of the determinant inequality in the first step gives the desired result. $\square$

We adapt an important concentration inequality for regularized least-squares estimates. Our proof follows the same basic strategy as Abbasi-Yadkori et al. (2011), but with modifications due to 1) the use of weighted least squares 2) the use of pseudo-rewards to estimate differential rewards 3) replacing the scaled diagonal regularization with Laplacian regularization.

**Lemma 7** (Adapted from Theorem 2 in Abbasi-Yadkori et al. (2011)). *For any $\delta > 0$, w.p. at least $1 - \delta$ the estimates $\{\hat{\Theta}_k\}_{k=0}^\infty$ in Algorithm 1 satisfies for any $\{x_k\}_{k=0}^\infty$,*

$$|x_k^\top(\hat{\Theta}_k - \Theta_k^*)| \leq \|x_k\|_{V_{k-1}^{-1}} \left(v_k\sqrt{2\log\left(\frac{\det(V_{k-1})^{1/2}\det(V_0)^{-1/2}}{\delta}\right)} + \lambda_{\max}(V_0)^{1/2}kB\right),$$

(10)

*where $v_k^2$ is the variance factor for the difference between the pseudo-reward and its mean at stage $k$. In particular, setting $x_k = V_{k-1}(\hat{\Theta}_{k-1} - \Theta_k^*)$ implies*

$$\|\hat{\Theta}_k - \Theta_k^*\| \leq v_k\sqrt{2\log\left(\frac{\det(V_{k-1})^{1/2}\det(V_0)^{-1/2}}{\delta}\right)} + \lambda_{\max}(V_0)^{1/2}kB$$

*holds w.p. at least $1 - \delta$ for all $k \geq 1$.*

*Proof.* Let $m_{i,t} = \tilde{\sigma}_{i,t}\phi(\mathbf{x}_{i,t})$ and $\rho_{i,t} = \tilde{\sigma}_{i,t}[\tilde{R}^f_{i,t}(s,\bar{a}) - \mathbb{E}[\tilde{R}^f_{i,t}|s,\bar{a}]]$. Further let

$$\xi_k \equiv \sum_{(i,t)\in\mathcal{O}_k} \tilde{\sigma}^2_{i,t}[\tilde{R}^f_{i,t}(s,\bar{a}) - \mathbb{E}[\tilde{R}^f_{i,t}|s,\bar{a}]]\phi(\mathbf{x}_{i,t})$$

$$= \sum_{(i,t)\in\mathcal{O}_k} \tilde{\sigma}_{i,t}[\tilde{R}^f_{i,t}(s,\bar{a}) - \mathbb{E}[\tilde{R}^f_{i,t}|s,\bar{a}]]m_{i,t}$$

$$= \sum_{(i,t)\in\mathcal{O}_k} m_{i,t}\rho_{i,t}$$

Then noting that $V_k = \sum_{(i,t)\in\mathcal{O}_k} m_{i,t}m^\top_{i,t} + V_0$ and letting $W_k$ be the diagonal matrix of weights $\tilde{\sigma}^2_{i,t}$, we have

$$\hat{\Theta}_k = V^{-1}_{k-1}b_k$$

$$= V^{-1}_{k-1}(\xi_k + \Phi^\top_k W_k \mathbb{E}[R^{\hat{f}_k}_k|\bar{a},s])$$

$$= V^{-1}_{k-1}\xi_k + V^{-1}_k \Phi^\top_k W_k \Delta_k \text{ by Lemma 2}$$

$$= V^{-1}_{k-1}\xi_k + V^{-1}_{k-1}\Phi^\top_k W_k \Phi_k \Theta^*_k$$

$$= V^{-1}_{k-1}\xi_k + V^{-1}_{k-1}(\Phi^\top_k W_k \Phi_k + V_0)\Theta^*_k - V^{-1}_{k-1}V_0\Theta^*_k$$

$$= V^{-1}_{k-1}\xi_k + \Theta^*_k - V^{-1}_{k-1}V_0\Theta^*_k$$

and thus

$$\hat{\Theta}_k - \Theta^*_k = V^{-1}_{k-1}(\xi_k - V_0\Theta^*_k)$$

which gives

$$|x^\top_k \hat{\Theta}_k - x^\top_k \Theta^*_k| \le \|x_k\|_{V^{-1}_{k-1}}(\|\xi_k\|_{V^{-1}_{k-1}} + \|V_0\Theta^*_k\|_{V^{-1}_{k-1}})$$

Now since $\xi_k$ is sub-Gaussian with variance factor $v^2_k$, by Theorem 1 in Abbasi-Yadkori et al. (2011), w.p. $1 - \delta$,

$$\|\xi_k\|^2_{V^{-1}_{k-1}} \le 2v^2_k \log\left(\frac{\det(V_{k-1})^{1/2}\det(V_0)^{-1/2}}{\delta}\right)$$

Further note

$$\|V_0\Theta^*_k\|^2_{V^{-1}_{k-1}} = \Theta^{*\top}_k V^\top_0 V^{-1}_{k-1} V_0\Theta^*_k$$

$$\le \|V^\top_0 V^{-1}_{k-1} V_0\|_2 \|\Theta^*_k\|^2_2$$

$$\le \|V_0\|^2_2 \|V^{-1}_{k-1}\|_2 \|\Theta^*_k\|^2_2$$

$$\le \lambda_{\max}(V_0)\|\Theta^*_k\|^2_2$$

$$\le \lambda_{\max}(V_0)k^2 B^2$$

and thus

$$|x^\top_k \hat{\Theta}_k - x^\top_k \Theta^*_k| \le \|x_k\|_{V^{-1}_{k-1}}\left(v_k\sqrt{2\log\left(\frac{\det(V^{-1}_{k-1})^{1/2}\det(V_0)^{-1/2}}{\delta}\right)} + \lambda_{\max}(V_0)^{1/2}kB\right)$$

**Corollary 2.** *If $\|f_{i,t} - r_{i,t}\|_u = \tilde{O}_P(k^{-1/4})$, then for any $\delta > 0$, there exists $C > 0$ s.t. w.p. at least $1 - \delta$ the estimates $\{\hat{\Theta}_k\}^\infty_{k=0}$ in Algorithm 1 satisfies for any $\{x_k\}^\infty_{k=0}$,*

$$|x^\top_k(\hat{\Theta}_k - \Theta^*_k)| \le \|x_k\|_{V^{-1}_{k-1}}\left(\left(\frac{C\log^{2m}(k)}{k^{1/2}} + \sigma^2 c^2\right)\sqrt{2\log\left(\frac{\det(V_{k-1})^{1/2}\det(V_0)^{-1/2}}{\delta/2}\right)} + \lambda_{\max}(V_0)^{1/2}kB\right),$$

$$(11)$$

*In particular, setting $x_k = V_{k-1}(\hat{\Theta}_{k-1} - \Theta^*_k)$ implies*

$$\|\hat{\Theta}_k - \Theta^*_k\| \le \left(\frac{C}{k^{1/2}} + \sigma^2 c^2\right)\sqrt{2\log\left(\frac{\det(V_{k-1})^{1/2}\det(V_0)^{-1/2}}{\delta/2}\right)} + \lambda_{\max}(V_0)^{1/2}kB$$

*holds w.p. at least $1 - \delta$ for all $k \ge 1$.*

*Proof.* Use Corollary 1 and Lemma 7, each with $\delta/2$. Then w.p. at least $1 - \delta$ the result holds. $\qquad\square$

$\qquad\square$

We next state a slightly modified form of a standard result of RLS (Lemma 11 in Abbasi-Yadkori et al. (2011)) that helps to guarantee that the prediction error is cumulatively small. This bounds the sum of quadratic forms where the matrix is the inverse Gram matrix and the arguments are the feature vectors. We use such terms to construct a martingale in the regret bound so that we can bound such terms and the martingale.

**Proposition 1.** *Let $\lambda \geq 1$ and $\gamma \geq 1$. For any arbitrary sequence $(\boldsymbol{x}_{i,t})_{(i,t)\in\mathcal{O}_k}$, let*

$$V_{k+1} \equiv \sum_{(i,t)\in\mathcal{O}_k} \tilde{\sigma}_{i,t}^2 \phi(\boldsymbol{x}_{i,t})\phi(\boldsymbol{x}_{i,t})^\top + V_0,$$

*be the regularized Gram matrix. Then*

$$\sum_{k=1}^{K} \sum_{(i,t)\in\mathcal{O}_k\setminus\mathcal{O}_{k-1}} \|\phi(\boldsymbol{x}_{i,t})\|_{V_k^{-1}}^2 \leq 2c \log\left(\frac{\det(V_{K+1})}{\det(V_0)}\right).$$

*where $c$ is a constant such that $0 < \frac{1}{c} < \tilde{\sigma}_{i,t}^2 \forall i, t \in \mathbb{N}$. Further,*

$$\log \det(V_{K+1}) \leq 2Kd \log\left(\left[\gamma + \lambda M + \frac{K+1}{8d}\right]\right)$$

*Proof.* By Lemma 11 in Abbasi-Yadkori et al. (2011), we have

$$\sum_{k=1}^{K} \sum_{(i,t)\in\mathcal{O}_k\setminus\mathcal{O}_{k-1}} \tilde{\sigma}_{i,t}^2 \|\phi(\boldsymbol{x}_{i,t})\|_{V_k^{-1}}^2 \leq 2 \log\left(\frac{\det(V_{K+1})}{\det(V_0)}\right).$$

The lower bound on the weights implies

$$\sum_{k=1}^{K} \sum_{(i,t)\in\mathcal{O}_k\setminus\mathcal{O}_{k-1}} \|\phi(\boldsymbol{x}_{i,t})\|_{V_k^{-1}}^2 \leq 2c \log\left(\frac{\det(V_{K+1})}{\det(V_0)}\right)$$

as desired.

By definition, we have $tr(\tilde{\sigma}_{i,t}^2 \phi(\boldsymbol{x}_{i,t})\phi(\boldsymbol{x}_{i,t})^\top) \leq 2/4$ since $\|x_{i,t}\| \leq 1$ and $\tilde{\sigma}_{i,t}^2 \leq (1/2)^2$; and $tr(L_\otimes) = tr(L) \cdot tr(I) = KMd$. Then by Hadamard's inequality we have

$$\begin{aligned}
\log \det(V_{K+1}) &\leq 2Kd \log\left(\frac{1}{2Kd} tr(V_{K+1})\right) \\
&= 2Kd \log\left(\frac{1}{2Kd}\left[\gamma 2Kd + \lambda 2KMd + \frac{2}{4}K \cdot (K+1)/2\right]\right) \\
&= 2Kd \log\left(\left[\gamma + \lambda M + \frac{K+1}{8d}\right]\right)
\end{aligned}$$

$\qquad\square$

Finally, we state Azuma's concentration inequality which describes concentration of super-martingales with bounded differences and is useful in controlling the regret due to the randomization of Thompson sampling.

**Proposition 2** (Azuma's concentration inequality)**.** *If a super-martingale $(Y_t)_{t\geq 0}$ corresponding to a filtration $\mathcal{F}_t$ satisfies $|Y_t - Y_{t-1}| < c_t$ some constant $c_t$ for all $t = 1, \ldots, T$ then for any $\alpha > 0$:*

$$P\left(Y_T - Y_0 \geq \alpha\right) \leq \exp\left(-\frac{\alpha^2}{2\sum_{t=1}^{T} c_t^2}\right).$$

### G.3 Proof of Theorem 1

The proof follows closely from Abeille & Lazaric (2017) with several adjustments. Assumption 2 implies that we only need to consider the unit ball $\mathcal{X} = \{\|x\| \leq 1\}$. Then the regret can be decomposed into

$$\underbrace{\sum_{k=1}^{K} \frac{1}{k} \sum_{(i,t) \in \mathcal{O}_k \setminus \mathcal{O}_{k-1}} \left( (\phi(x_{i,t}^\star))^\top \Theta_k^\star - \phi(x_{i,t})^\top \tilde{\Theta}_k \right)}_{R^{TS}(K)} + \underbrace{\sum_{k=1}^{K} \frac{1}{k} \sum_{(i,t) \in \mathcal{O}_k \setminus \mathcal{O}_{k-1}} \left( \phi(x_{i,t})^\top \tilde{\Theta}_k - \phi(x_{i,t})^\top \Theta_k^\star \right)}_{R^{RLS}(K)}$$

where $\phi(x_{i,t}^\star)$ is the context vector under the optimal action and $\Theta_k^\star$ is the true parameter value. The first term is the regret due to the random deviations caused by sampling $\tilde{\Theta}_k$ and whether it provides sufficient useful information about the true parameter $\Theta_k^\star$. The second term is the concentration of the sampled term around the true linear model for the advantage function.

**Definition 2.** *We define the filtration $\mathcal{F}_k$ as the information accumulated up to stage $k$ before the sampling procedure, that is, $\mathcal{F}_k = (\mathcal{F}_1, \sigma(x_1, r_2, x_2, \ldots, x_{k-1}, r_{k-1}))$, and filtration $\mathcal{F}_k^x$ as the information accumulated up to stage $k$ and including the sampled context, that is, $\mathcal{F}_t = (\mathcal{F}_1, \sigma(x_1, r_2, x_2, \ldots, x_{k-1}, r_{k-1}, x_k))$.*

**Bounding $R^{RLS}(T)$.** We decompose the second term into the variation of the point estimate and the variation of the random sample around the point estimate:

$$\sum_{k=1}^{K} \frac{1}{k} \sum_{(i,t) \in \mathcal{O}_k \setminus \mathcal{O}_{k-1}} \left( \phi(x_{i,t})^\top \tilde{\Theta}_k - \phi(x_{i,t})^\top \hat{\Theta}_k \right) + \sum_{k=1}^{K} \frac{1}{k} \sum_{(i,t) \in \mathcal{O}_k \setminus \mathcal{O}_{k-1}} \left( \phi(x_{i,t})^\top \hat{\Theta}_k - \phi(x_{i,t})^\top \Theta_k^\star \right)$$

The first term describes the deviation of the TS linear predictor from the RLS one, while the second term describes the deviation of the RLS linear predictor from the true linear predictor. The first term is controlled by the construction of the sampling distribution $D^{TS}$, while the second term is controlled by the RLS estimate being a minimizer of the regularized cumulative squared error in (5). In particular, the first term will be small when the TS estimate concentrates around the RLS one, while the second will be small when the RLS estimate concentrates around the true parameter vector. The next proposition gives a lower bound on the probability that, for all stages, both the RLS parameter vector concentrates around the true parameter vector and the TS parameter vector concentrates around the RLS one.

Recall that

$$\beta_k(\delta) = v_k \left[ 2 \log \left( \frac{\det(V_k)^{1/2}}{\det(V_0)^{1/2} \delta/2} \right) \right]^{1/2} + B$$

**Proposition 3.** *Let $\hat{E}_k$ denote the event that $\hat{\Theta}_k$ concentrates around the true parameter for all $l \leq k$, i.e., $\hat{E}_k = \{\forall l \leq k, \|\hat{\Theta}_l - \Theta_l^\star\|_{V_l} \leq \beta_l(\delta')\}$. Let $\gamma_k(\delta) \equiv \beta_k(\delta') \sqrt{cd \log \frac{c'd}{\delta}}$ Let $\tilde{E}_k$ denote the event that $\tilde{\Theta}_l$ concentrates around the estimated parameter for all $l \leq k$, i.e., $\tilde{E}_k = \{\forall l \leq k, \|\tilde{\Theta}_l - \hat{\Theta}_l\|_{V_l} \leq \gamma_l(\delta')\}$. Let $E_k = \hat{E}_k \cap \tilde{E}_k$. Then $P(E_k) \geq 1 - \delta/2$.*

*Proof.* Let $\delta' = \delta/4K$, then Lemma 7 and a union bound give us

$$P(\hat{E}_K) = P(\cap_{k=1}^{K} \{\|\Theta_k - \Theta^\star\|_{V_k} \leq \beta_k(\delta')\})$$

$$= 1 - \sum_{k=1}^{K} P(\|\Theta_k - \Theta^\star\|_{V_k} > \beta_k(\delta'))$$

$$= 1 - \sum_{k=1}^{K} \delta' = 1 - \delta'K = 1 - \delta/4.$$

Applying the TS sampling distribution and $\tilde{\Theta}_k = \hat{\Theta}_k + \beta_k(\delta')V_k^{-1/2}\eta_k$ where $\eta_t$ is drawn i.i.d. from $D^{TS}$ we have

$$P\left(\|\tilde{\Theta}_k - \hat{\Theta}_k\|_{V_k} \leq \beta_k(\delta')\sqrt{cd\log\left(\frac{c'd}{\delta'}\right)}\right) = P\left(\|\eta_k\| \leq \sqrt{cd\log\left(\frac{c'd}{\delta'}\right)}\right) \geq 1 - \delta'.$$

by Definition 1. A union-bound argument yields the conclusion. $\qquad\square$

We can then bound $R^{RLS}(K)$ by leveraging Lemma 7 and decomposing the error via

$$R^{RLS}(K) \leq \sum_{k=1}^{K} \frac{1[E_K]}{k}\left[\sum_{(i,t)\in\mathcal{O}_k\backslash\mathcal{O}_{k-1}} |\phi(x_{i,t})^\top(\tilde{\Theta}_k - \hat{\Theta}_k)|\right]$$
$$+ \sum_{k=1}^{K} \frac{1[E_K]}{k}\left[\sum_{(i,t)\in\mathcal{O}_k\backslash\mathcal{O}_{k-1}} |\phi(x_{i,t})^\top(\hat{\Theta}_k - \Theta_k^*)|\right]$$

By definition of the event $E_K$, we have

$$|\phi(x_{i,t})^\top(\tilde{\Theta}_k-\hat{\Theta}_k)|1[E_k] \leq \|\phi(x_{i,t})\|_{V_k^{-1}}\gamma_k(\delta'), \quad |\phi(x_{i,t})^\top(\hat{\Theta}_k-\Theta^\star)|1[E_k] \leq \|\phi(x_{i,t})\|_{V_k^{-1}}\beta_k(\delta')$$

so from Proposition 1, we have

$$\sum_{k=1}^{K}\frac{1[E_K]}{k}\left[\sum_{(i,t)\in\mathcal{O}_k\backslash\mathcal{O}_{k-1}} |\phi(x_{i,t})^\top(\tilde{\Theta}_k - \hat{\Theta}_k)|\right]$$
$$\leq \gamma_K(\delta')\sum_{k=1}^{K}\left[\sum_{(i,t)\in\mathcal{O}_k\backslash\mathcal{O}_{k-1}} \frac{1}{k}\|\phi(x_{i,t})\|_{V_k^{-1}}\right]$$
$$\leq \gamma_K(\delta')\sqrt{\sum_{k=1}^{K}\sum_{(i,t)\in\mathcal{O}_k\backslash\mathcal{O}_{k-1}} \frac{1}{k^2}}\sqrt{\sum_{k=1}^{K}\sum_{(i,t)\in\mathcal{O}_k\backslash\mathcal{O}_{k-1}} \|\phi(x_{i,t})\|_{V_k^{-1}}^2}$$
$$\leq \gamma_K(\delta')\sqrt{\sum_{k=1}^{K}\frac{1}{k}}\sqrt{\sum_{k=1}^{K}\sum_{(i,t)\in\mathcal{O}_k\backslash\mathcal{O}_{k-1}} \|\phi(x_{i,t})\|_{V_k^{-1}}^2}$$
$$\leq \gamma_K(\delta')\sqrt{H_K}\sqrt{\sum_{(i,t)\in\mathcal{O}_k} \|\phi(x_{i,t})\|_{V_k^{-1}}^2}$$
$$\leq \gamma_K(\delta')\sqrt{H_K}\sqrt{2c\log\left(\frac{\det(V_{K+1})}{\det(V_0)}\right)}.$$

Using a similar derivation for the $\beta_k(\delta')$ case, we obtain

$$R^{RLS}(K) \leq (\beta_K(\delta') + \gamma_K(\delta'))\sqrt{\sum_{k=1}^{K}\sum_{(i,t)\in\mathcal{O}_k\backslash\mathcal{O}_{k-1}} \frac{1}{k^2}}\sqrt{2c\log\left(\frac{\det(V_{K+1})}{\det(V_0)}\right)}$$
$$\leq (\beta_K(\delta') + \gamma_K(\delta'))\sqrt{\sum_{k=1}^{K}\frac{1}{k}}\sqrt{2c\log\left(\frac{\det(V_{K+1})}{\det(V_0)}\right)}$$
$$\leq (\beta_K(\delta') + \gamma_K(\delta'))\sqrt{H_K}\sqrt{2c\left[2Kd\log\left(\gamma + \lambda M + \frac{K+1}{8d}\right) - \log\det(V_0)\right]}$$

with probability at least $1 - \delta/2$ by Proposition 3, where $H_K$ is the harmonic number. Note that $H_K \sim \log(K)$ for large $K$.

**Bounding $R^{TS}(T)$.** Leveraging Abeille & Lazaric (2017), Definition 1 lets us bound $R^{TS}(K)$ under the event $E_k$ by

$$R^{TS}(K) \le \sum_{k=1}^{K} \frac{1}{k} R_k^{TS} 1[E_k] \le \frac{4\gamma_K(\delta')}{d} \sum_{k=1}^{K} \frac{1}{k} \sum_{(i,t)\in\mathcal{O}_k\backslash\mathcal{O}_{k-1}} \mathbb{E}\left[\|\phi(x_{i,t}^\star)(\tilde{\Theta})\|_{V_k^{-1}}|\mathcal{F}_k\right] \quad (12)$$

We re-write the sum in (12) as:

$$\sum_{k=1}^{K} \frac{1}{k} \sum_{(i,t)\in\mathcal{O}_k\backslash\mathcal{O}_{k-1}} \|\phi(x_{i,t})\|_{V_k^{-1}} + \underbrace{\sum_{k=1}^{K} \frac{1}{k} \sum_{(i,t)\in\mathcal{O}_k\backslash\mathcal{O}_{k-1}} \left(\mathbb{E}\left[\|\phi(x_{i,t}^\star)(\tilde{\Theta})\|_{V_k^{-1}}|\mathcal{F}_k\right] - \|\phi(x_{i,t})\|_{V_k^{-1}}\right)}_{R_2^{TS}}$$

The first term is bounded by Proposition 1:

$$\sum_{k=1}^{K} \frac{1}{k} \sum_{(i,t)\in\mathcal{O}_k\backslash\mathcal{O}_{k-1}} \|\phi(x_{i,t})\|_{V_k^{-1}} \le \sqrt{2cH_K \log\left(\frac{\det(V_{K+1})}{\det(V_0)}\right)}$$

The second term is a martingale by construction and so we can apply Azuma's inequality. Under Assumption 2, so since $V_k \le \frac{1}{\lambda}I$ we have

$$\mathbb{E}\left[\|\phi(x_{i,t})^\star(\tilde{\Theta})\|_{V_k^{-1}}|\mathcal{F}_t\right] - \|\phi(x_{i,t})\|_{V_k^{-1}} \le \frac{2}{\sqrt{\lambda}}, \quad a.s.$$

This provides the upper-bound

$$R^{TS}(K) \le \frac{4\gamma_K(\delta')}{d}\left(\sqrt{\frac{8K}{\lambda}\log\left(\frac{4}{\delta}\right)} + \sqrt{4cH_K Kd \log\left(\gamma + \lambda M + \frac{K+1}{8d}\right) - \log\det(V_0)}\right).$$

**Overall bound.** Putting together the two bounds under a union bound argument yields the upper bound in Theorem 1; specifically, we have

$$\left(\beta_K(\delta') + \gamma_K(\delta')\left[1 + \frac{4}{d}\right]\right)\sqrt{4cH_K Kd \log\left(\gamma + \lambda M + \frac{K+1}{8d}\right) - \log\det(V_0)}$$

$$+ \frac{4\gamma_K(\delta')}{p}\sqrt{\frac{8K}{\lambda}\log\left(\frac{4}{\delta}\right)}$$