# OpenReview forum: "Debiased Machine Learning and Network Cohesion for Doubly-Robust Differential Reward Models in Contextual Bandits"
_ICLR.cc/2024/Conference — Submitted to ICLR 2024_

### Official Review · Reviewer_iXa9 · 2023-10-25

**Soundness:** 2 fair
**Presentation:** 2 fair
**Contribution:** 2 fair
**Rating:** 5
**Confidence:** 3

**Summary:**

This paper studies mobile health intervention policies. Specifically, the setting that has the following four features, 1) the time-varying nature of feedback (captured by the hidden parameter theta^*_t) 2) the Non-linear relationship between action and outcomes (captured by conditional model for the observed reward in eq(1)) 3) Intervention efficacy changing over time and 4) similar features lead to similar outcomes (captured through the user graph). To tackle this they introduce a novel reward model in eq(1) that mimics a linear differential equation with a potentially non-linear component g_t(s) (which they consider as the baseline reward). Finally, to capture the idea that users showing similar symptoms require similar intervention they introduce the user graph. The assumption is that connected users share similar underlying vectors \theta_i implying that the rewards received from one user can provide insights into the behavior of other connected users. They also introduce a time connection graph G_t (the use of which is not fully clear to me). Finally, they propose an algorithm that has access to these graphs and proceeds in stages. In each stage k, it puts the data into partitions and uses an MAB algorithm (TS here) to select actions from these partitions, uses a least square estimate to the function f_hat(m), observes the reward, and updates the model. This approach balances exploration and exploitation simultaneously. They theoretically analyze their algorithm with a regret upper bound and conduct empirical evaluations.

**Strengths:**

1) The setting seems novel and relevant. However, the reward model needs more justification. See questions
2) The proposed algorithm is theoretically analyzed with a regret upper bound.
3) They conduct experiments to justify their algorithm.

**Weaknesses:**

1) The paper presentation can be improved. Some of the notations are never introduced. For example \delta in eq (1) is never introduced. In section 3.2 I think you should clearly state this as an assumption that connected users in graph share similar \theta_i.
2) The model requires more justification. See the question section below.
3) The technical novelty seems not that significant.
4) Needs more experiments to justify the setting.

**Questions:**

1) How is the graph G_time used? You suddenly introduce it in section 4 while discussing the algorithm. Do you construct a similar Laplacian matrix L and calculate tr(\Theta^\top L \Theta)?
2) Why is Thompson sampling here and not any other MAB algorithm? From the pseudocode, or even from the regret proof overview (I did not check the appendix) it did not seem to me that TS is giving any benefit. Can you elaborate on this?
3) Regarding the feedback model, the baseline reward $g_t(s)$ is suddenly introduced. What is it and why is it significant? Also, a follow-up is similar to conservative bandits studied in the literature, where the reward must not go down below the baseline reward. Is that work somehow connected to your setting?
4) I was slightly confused with the regret definition. First, is the observed set \O_k is just the history in stage k? Secondly, why are you averaging over each of the stages k? The usual definition of regret will be summing over all stages of $k\in [K]$?
5) Regarding the algorithm design I have a few questions. Observe that at the beginning of every stage k, you randomly assign the points to a partition in m. It might happen that in the first partition itself, you end up with a sequence of highly non-informative samples. Then when you use your MAB algorithm (TS here) to select an action you might end up with a very bad estimation of f_hat(m). How do you guard against this possibility? Also in the same line, how do you ensure sufficient diversity while allocating the points to partition? Do you have any underlying assumptions on this?
6) It is not clear how the regret bound is analyzed. What is the key technical challenge in the analysis of your regret bound and how it differs from Abeille & Lazaric (2017)? Also, how does the harmonic number show up? It seems the regret bound scales as $\sqrt{K}$ which matches the standard linear bandit scaling of $\sqrt{T}$. However, where does the dimension d show up?

**Details Of Ethics Concerns:**

Not applicable.

---

> ### Author Response · Authors · 2023-11-23
>
> Thank you for your feedback: we address your mentioned weaknesses and questions.
>
> ## Weaknesses
> 1. Thank you, we have revised the paper to incorporate these.
> 2. The primary technical contribution is the use of double machine learning. By explicitly modeling the baseline reward, the sub-Gaussian variance factor converges to terms that depend only on the model noise and propensity weights, but where the dependence on the mean reward goes to $0$ with the number of stages.
>
> ## Questions
> 1. Correct.
> 2. The theory behind DML requires a condition called "overlap" where the treatment assignment probabilities have to be bounded away from zero and one. This rules out deterministic rule selection (e.g., UCB), so Thompson sampling is a clear choice. Furthermore, in mobile health, it is common to analyze the study results after applying a bandit algorithm. [1] describes the standard analysis. The goal is to estimate "causal excursion effects." These effects quantify what would happen if we followed a policy up until time $t$ and then deviated from it. Estimating them also requires overlap. Otherwise, we would encounter scenarios in which we have no data to estimate deviations from the policy because the policy is followed deterministically.
> 3. We apologize for lacking clarification on the terms we used, and we added the following to the updated manuscript:  "The term baseline reward' refers to the reward observed under arm zero, where individuals are randomized to not receive any treatment." Modeling this baseline reward is crucial because it has the potential to substantially decrease the size of resulting confidence sets for the causal parameter of interest. This, in turn, enhances the precision of information available for decision-making. Intuitively, this reduction occurs because the signal-to-noise ratio effectively decreases; while the signal remains consistently estimated, the residual variance becomes smaller with an accurate baseline model. In response to your follow-up question, it's important to note that the setting we are considering here is not particularly closely related to conservative bandits. In our context, the "baseline reward" is defined under one of the multiple treatment options rather than representing the "worst-case scenario". Consequently, under certain circumstances, it is reasonable to expect a higher reward under no treatment allocation.
> 4. No quite, we consider the history up until stage k.
> 5. We agree that this requires further clarification. With sample splitting, there is a chance that the random assignment may lead to reduced predictive capabilities of any single baseline reward model.  When building predictions for a particular fold, the training set contains all other samples. Thus, a single split is unlikely to dominate the prediction performance. As the stages grow in size, the number of samples per split is large enough that the issue is not seen empirically in our synthetic and real-world experiments. In the main text, we now re-emphasize the alternative approach of Option 2 where we update predictions in an online manner.
> 6. The technical challenges are: first, the confidence ellipsoids depend on the sub-Gaussian variance factor of the pseudo-reward and need to be derived for the DML pseudo-reward. Second, two results in (Abbassi-Yadkori et al. 2011) need to be re-proved: the first is Lemma 7, the linear predictor bound used as a key step to derive the final regret bounded. It requires handling the fact that our regularized least squares estimate now uses our DML pseudo-reward instead of the observed reward. The second, proposition $1$, requires care as we are doing *weighted* regularized least squares. The original version required by (Abeille et al. 2017) requires an upper bound on the sum of squared norms, but applying (Abbassi-Yadkori et al. 2011) only gives us an upper bound on the sum of weighted squared norms. In order to derive the needed bound, we use a standard mobile health assumption that the probability of no treatment is upper and lower bounded (Greenewald et al. 2017) and then apply (Abbassi-Yadkori et al. 2011) to obtain our version of their results that have an upper bound that depends on $c>0$, the lower bound on the weights. Finally, the regret bound itself needs to handle the fact that we have stages with multiple individuals (increasing by one) per stage. This leads to a sum over stages and participants within stages of the difference between RLS and TS (and RLS and true) linear predictors. By some manipulation and an application of Cauchy Schwartz, we see a sum of $\frac{1}{k}$ over stages, which leads to the harmonic number. The dimensionality shows up as the sum of squared norms is bounded by a log ratio of determinants, which is bounded by terms involving the dimensionality.
>
> [1] Boruvka, Audrey, et al. "Assessing time-varying causal effect moderation in mobile health." Journal of the American Statistical Association 113.523 (2018): 1112-1121.

---

### Official Review · Reviewer_isfh · 2023-10-26

**Soundness:** 2 fair
**Presentation:** 1 poor
**Contribution:** 2 fair
**Rating:** 6
**Confidence:** 3

**Summary:**

This paper presents a double/debiased machine learning (DML)-based nonlinear contextual bandit algorithm that takes into account network cohesion.

**Strengths:**

1. This paper thoroughly reviewed previous literature.
2. The simulation study has been thoroughly conducted, which enhances the empirical power of the proposed method and makes it more plausible.

**Weaknesses:**

Overall, this paper is overly biased towards theoretical aspects. I understand “what” problems authors have solved, but I cannot understand “why” the authors have solved the problem.

1. The storyline for the Introduction is somewhat drastic. Without any examples of the contextual bandit on mHealth, it's difficult to understand what sets mHealth apart and makes it special.
2. The motivation is quite weak in this paper. I understand that this paper relaxes many assumptions made in previous literature. However, there is a lack of discussion regarding the practical benefits of relaxing these assumptions and addressing complex problems. It is important to explain the significance of relaxing the assumptions/challenges (1,2,3,4) mentioned in the first paragraph and provide examples of how each of the four challenges mentioned in the Introduction hampers the practicality of mHealth. Without such examples, it is difficult to persuade readers why "DML-TS-NNR" is necessary.
3. Please cite Chernozhukov et al., (2018) when mentioning DML.
4. In the related work section, the paper does not mention the Bandit literature that takes into account network cohesion. I’d like to see the comparison between this paper versus the Bandit literature taking account of the network interference.
5. I believe that the related work section does not include any papers on DML (or doubly robust) contextual bandits. Please correct me if I am mistaken.
6. The math wall of this paper is huge. The notations are heavy, but no verbal explanation is provided. For example, in Theorem 1, I can't pull the implication of this regret bound. Is it good or bad?

**Questions:**

1. This paper is written in a way that highlights how the proposed method is best suited for mHealth. What is the reason for this?
2. To my knowledge, the paper doesn’t assume the iid. However, the DML theory with sample splitting is developed based on the iid assumption. So, there must be more explanation of how this iid-based method can be used to address the problems in non-iid settings. Also, is the method provided by Chen et al., (2022) applicable to the non-iid setting?
3. This paper used a DML-based method. Then, how can I read the fast-convergence properties or doubly robustness of the algorithm in Theorem 1?

---

> ### Author Response · Authors · 2023-11-23
> **Addressing Weaknesses**
>
> Thank you for your helpful feedback. We address your comments below.
>
> ## Weaknesses
>
> 1. We realize that the introduction may not sufficiently establish the connection between mHealth contextual bandits. We have incorporated the following paragraph into the updated manuscript: "Mobile health (mHealth) and contextual bandit algorithms share a close connection in the realm of personalized healthcare interventions. mHealth leverages mobile devices to deliver health-related services, making it a powerful tool for real-time monitoring and intervention. Contextual bandit algorithms, on the other hand, are a class of machine learning techniques designed to optimize decision-making in situations where actions have contextual dependencies. The synergy arises when mHealth applications deploy contextual bandit algorithms to tailor interventions based on individual health data and context. For example, in a mobile health setting, a contextual bandit algorithm might dynamically adapt the type and timing of health-related notifications or interventions based on the user's current health status, historical behavior, and contextual factors like location or time of day.'
> 2. We restate the list here: "bandit algorithms must account for (1) the time-varying nature of the outcome variable, (2) nonlinear relationships between states and outcomes, (3) the potential for intervention efficacy to change over time
> (due, for instance, to habituation as in Psihogios et al. (2019)), and (4) the fact that similar participants tend to respond similarly to interventions (Künzler et al., 2019)." For (1), an example would be that step counts and activity more generally could depend on weather, even when no action is taken [1]. Explicitly modeling the baseline via DML would incorporate this into our model. For (2) the activity dependence on the weather may be non-linear. Using DML allows us to use a non-linear baseline. For (3), we cited an example, but it relates to our penalization encouraging similar effects for similar time points. (4) relates to our penalization encouraging similar effects for similar users.
> 3. We cite it on the top of page 3, but we agree that it would be helpful to cite it at its earliest reference in the main paper (top of page 2).
> 4. We mention these papers in the graph bandits section. These use a graph that encourages similarity between similar users, while our approach encourages similarity both between similar users and similar time points.
> 5. Thank you for pointing this out. Other papers have used doubly robust estimators, but for slightly different purposes. We use a doubly robust estimator that allows us to debias and lower variance from explicitly modeling the mean reward including the baseline. Other papers [2, 3] use a doubly robust estimator to obtain novel regret bounds in the linear and generalized linear settings, respectively. However, they do not treat the mHealth setting where there is the possibility of taking no action. We will add these to our related work section.
> 6. We thank the reviewer for raising this concern and apologize for the amount of notation.  To address the heavy amounts of notation, we will add a notation glossary to the appendix.  As Theorem 1 is important for understanding performance of our algorithm, we have added Remark 1.  Below find the exact language added.
>
> **Remark 1**
> Observe that the regret bound is sublinear in the number of stages and scales only with the complexity of the differential reward $d$, not the complexity of the baseline reward $g$. As the second term scales with $\sqrt{K}$, for large $K$ we can focus on the first term which scales $O \left( \sqrt{c \cdot d \cdot \log^2(K) K} \right)$.  Prior work scales sublinearly with the number of decision times $T$ as they assume either a single contextual bandit ($n=1$) or a fixed number of individuals~$n$. Action centering scales as $O(d^2 \sqrt{T \log(T)})$ while Graph Bandits scale as $O(\sqrt{\tilde d n T \log(T)})$ where $\tilde d$ is the complexity of the joint baseline and differential reward model.  Interestingly, at stage $K$ we see $n=K$ individuals over $n=K$ decision times (with different number of observations per individual); however, we do not see a regret scale with $\sqrt{nT} = K$. Instead we only receive an extra $\log(K)$ factor reflecting the benefit of pooling on average regret.
>
> [1] Turrisi, Taylor B., et al. "Seasons, weather, and device-measured movement behaviors: a scoping review from 2006 to 2020." International Journal of Behavioral Nutrition and Physical Activity 18 (2021): 1-26.
> [2] Kim, Wonyoung, Gi-Soo Kim, and Myunghee Cho Paik. "Doubly robust thompson sampling with linear payoffs." Advances in Neural Information Processing Systems 34 (2021): 15830-15840.
> [3] Kim, Wonyoung, Kyungbok Lee, and Myunghee Cho Paik. "Double doubly robust thompson sampling for generalized linear contextual bandits." Proceedings of the AAAI Conference on Artificial Intelligence. Vol. 37. No. 7. 2023.

---

> ### Author Response · Authors · 2023-11-23
> **Addressing Questions**
>
> ## Questions
> 1. There are two important characteristics of mHealth settings: 1) The combination of having a baseline `do nothing' action and a non-linear baseline reward (e.g. context outside of the bandit action may affect step count, such as the weather) (Greenewald et al. 2017). Making too many recommendations can lead to burden and habituation for users, and thus in mHealth we would like to not make recommendations at every time step. Our DML based approach allows us to explicitly model the baseline rewards (e.g. how weather affects step count independent of action). 2) Time-varying effects. For instance, at the beginning of a study participants may be more receptive to recommendtations than towards the end. By including a penalty that encourages similarity between time points in addition to the standard similarity between users, we can model these time-varying effects.
> 2. Thank you for bringing up this concern. In the main text on Page 4, we outlined two approaches for learning the conditional expectation model $f_{t}(s,a)$. Option 1 involves sample splitting across time and participants, assuming additive i.i.d. errors and no delayed or spill-over effects. This assumption is plausible in the mHealth setting, where we do not anticipate an adversarial environment. Option 2 relaxes the i.i.d. error assumption by updating $f_{t}(s,a)$ using observed history data in an online fashion, keeping estimated pseudo-outcomes fixed at each stage. Consequently, the new observations become conditionally independent of the history data. In this context, we can leverage subsampling techniques introduced in (Chen et al. 2022) as an alternative method to estimate pseudo-outcomes, avoiding the need for sample splitting.
> 3. The main contribution for DML in Theorem 1 is in the $\beta_K$ and $\gamma_K$ terms, which represent the regularized least squares and Thompson sampling high probability ellipsoids, respectively. Each of these depend on the sub-Gaussian variance factor, which is smaller due to using the doubly robust DML pseudo-reward. In the setting of (Greenewald et al. 2017), this variance factor always depends on the mean reward. In our setting, the terms involving the mean reward go to $0$ with the number of stages.

---

### Official Review · Reviewer_HoU5 · 2023-11-01

**Soundness:** 3 good
**Presentation:** 2 fair
**Contribution:** 2 fair
**Rating:** 5
**Confidence:** 3

**Summary:**

This paper studied the problem of learning mobile health (mHealth) intervention policies. The paper proposed a new Thompson sampling
based algorithm, named “DML-TS-NNR”, which achieves two desirable features: (1) pool information across individuals and time (2)  model the differential reward linear model with a time-varying baseline reward. Theoretical guarantees on the pseudo-regret are provided, and are supported by empirical evaluations.

**Strengths:**

- The paper studied a significant and practical problem motivated by mHealth applications, and overall the model, proposed algorithm and its analysis were presented with a good clarity.
-  The proposed DML-TS-NNR algorithm achieves reduced confidence set sizes and provides an improved high probability regret bound.
- The algorithm design takes the motivating application into account: the individuals’ enrollment occurs in a staggered manner, which mimics the recruitment process in mHealth studies.
- Empirical experiments support the theoretical result and show that the proposed algorithm outperforms other baselines.

**Weaknesses:**

- This paper is not presented in the correct ICLR latex template format
- Section presents the basic model, however the connection between the model and the particular mHealth application was missing, especially how the model and its assumption fit into the application was not elaborated clearly.
- It was unclear the major innovation of the proposed algorithm in comparison to related prior works, e.g. (Yang et al., 2020; Tomkins et al., 2021).

**Questions:**

- Can the algorithm be extended when the network structure is not fully known?

---

> ### Author Response · Authors · 2023-11-23
>
> Thank you for your helpful feedback. We address your listed weaknesses and comments.
>
> ## Weaknesses
>
> * We appreciate the feedback. We noticed that there was a bolding issue for the title, which we have fixed. If there are other specific issues you notice, we would be happy to address them.
> * We restate the list of mHealth concerns here and connect them to our model: "Bandit algorithms must account for (1) the time-varying nature of the outcome variable, (2) nonlinear relationships between states and outcomes, (3) the potential for intervention
> efficacy to change over time (due, for instance, to habituation as in Psihogios et al. (2019)), and (4) the fact that similar participants tend to respond similarly to interventions (Kunzler et al., 2019)." For (1), an example would be that step counts and activity more generally could depend on weather, even when no action is taken (T. B. Turrisi et al. 2021). Explicitly modeling the baseline via DML would incorporate this into our model. For (2) the activity dependence on the weather may be non-linear. Using DML allows us to use a non-linear baseline. For (3), we cited an example, but it relates to our penalization encouraging similar effects for similar time points. (4)
> relates to our penalization encouraging similar effects for similar users.
>
> * Our major innovations are explicitly modeling the non-linear baseline rewards using double machine learning and encouraging similarity across time points in addition to across users. Yang et al. 2020 showed how to share information across similar users, but do not handle baseline rewards or sharing information across similar times. Tomkins et al. do pooling across users and time, but only allow for a linear baseline reward (we mention this in the related work section). Our DML approach reduces the variance of the sub-Gaussian variance factor that comes up in the regret bound through the terms $\beta_K$ and $\gamma_K$, which are the high probability confidence ellipsoid bounds for the regularized least squares and Thompson sampling estimators, respectively. The DML approach shrinks the upper bound on the size of these ellipsoids.
>
> ## Questions
> * This is important future work: thanks for mentioning this. However, existing approaches such as (Yang et al. 2020) make the same assumption of a known graph. We will work on this going forward.

---

### Official Review · Reviewer_V6y9 · 2023-11-04

**Soundness:** 4 excellent
**Presentation:** 4 excellent
**Contribution:** 3 good
**Rating:** 6
**Confidence:** 2

**Summary:**

This paper studies Thompson sampling in contextual bandit models applied for mHealth intervention applications. In existing works, the approaches focus on pooling information across individuals but not across time. Furthermore, in existing works, the baseline reward model is not modeled, limiting the ability to estimate the parameters in the differential reward model.

Towards overcoming these limitations, the paper proposes a new algorithm (called DML-TS-NNR) that achieves both of the above considerations. It considers nearest neighbors to allow pooling information across both, individual users and time. The algorithm also leverages the Double Machine Learning (DML) framework to model baseline rewards. This algorithm enables achieving improved statistical precision and can learn contextually-tailored mHealth intervention policies better.

Finally, the paper presents a theoretical analysis and provides regret guarantees. The paper also presents experimental studies on simulation datasets; experimental results show that the proposed approach displays robustness to potential misspecifications in the baseline reward model.

**Strengths:**

– Paper is well written; key contexts and essential background is well established.

– Related work: Discussion of related work is well organized and seems to  adequately cover background literature. Relevant papers also seem to be adequate cited elsewhere in the paper wherever useful.

– I think one of the main strengths is that the paper is theoretically grounded. The paper presents useful theoretical results on regret analysis of the algorithm proposed and clearly states the assumptions involved.

**Weaknesses:**

– Experiments:

1. While the heatmap establishes statistical significance (Fig 2, right), the actual difference (or lift/improvement provided) in Fig 2(left) seems marginal.

2. The empirical results in the main paper are from simulation studies. The paper also claims that it performs analysis on a real-world IHS dataset but that is deferred to the appendix.

3. Minor comment: Are error bars available for Fig 1? (Or are they so small that the bars are negligible given that it was run for 50 trials?)

**Questions:**

– The experiments section discussed three variants of the proposed method. Empirical results show two of these turn out to be best-performing (sec 5.1). So why is the third method relevant? And how do we decide which of the method is the best?

–  The simulation results In Fig 1 seem to suggest a sharp difference in cumulative regret of the methods tested. However, on the Valentine study (Fig 2), the differences between the cumulative rewards seem rather small and not as dramatic as Fig 1 results. Why is there such a difference?

**Details Of Ethics Concerns:**

N.A.

---

> ### Author Response · Authors · 2023-11-22
>
> Thank you for your helpful comments and positive assessment. We address your listed weaknesses and comments.
>
> Weaknesses:
> * In Figure 2 (left), the Valentine study outcome, namely step
> count, is presented after a log transformation. Therefore, the small values in the figure
> represent the difference between two log step counts. For instance, the DML-TS-NNR-RF
> algorithm achieves an average 0.035 relative expected reward. This implies that, after im-
> plementing our proposed algorithm, the step counts increased by $$\exp(0.035) − 1 = 3.5\%$$ more
> than what the constant randomization policy achieved. To provide further context, this
> translates to an additional 35 steps on top of the original 1000 steps under the constant
> randomization scheme within the hour following the randomized mobile notification. In the
> context of mobile health applications and our emphasis on the step count in the following hour as the outcome of interest, it’s typical for effect sizes to be on the smaller side.
> Considering this, the 3.5% increase should be seen as a positive indicator of improvement,
> highlighting the strong performance of our proposed method compared to other competing
> algorithms.
> * In conjunction with the simulation study, the main text
> focuses on one of the real-world data applications—the Valentine Study. The results of this
> case study provide compelling evidence that our proposed algorithm attains significantly
> higher average rewards compared to the competing algorithms. Given the difference in the
> outcome of interest, randomization policy, participants’ enrollment scheme, and the graph
> construction between the IHS study and the Valentine study, we have chosen to defer the
> discussion of the IHS study results to the appendix for clarity.
> * We agree that assessing the uncertainty is important. Since the error bars would be difficult
> to read in Figure 1 because there are so many methods, we instead performed pairwise
> comparisons regarding final (stage-125) regret. Table 1 of Appendix D shows the results.
> In the heterogeneous and nonlinear simulation settings (the challenging ones), our proposed
> methods achieve a significantly lower regret than the other methods.
>
> Comments:
> * In addition to testing whether our method worked well, we were also interested in evaluating
> how different aspects of the method contributed to its performance; does our method
> perform well due to (1) user pooling, (2) time pooling, (3) network cohesion, or (4) flexible
> modeling of the baseline? Including DML-TS-SU-BT allowed us to assess the contribution
> of user pooling. The results indicate that user pooling is beneficial (see the results for the
> nonlinear setting)—though, perhaps slightly less so than we expected.
> In practice, we recommend employing a version of our method that has distinct parameters
> for each user (such as DML-TS-NNR-BLM or DML-TS-NNR-BT). We recommend selecting
> a supervised learning algorithm that (1) can be quickly updated in an online fashion (2)
> performs well in cross-validation and (3) has low sample complexity. When data is being
> collected in an online manner, the supervised learning method could periodically be re-
> evaluated via cross-validation.
> * Thank you for bringing up this concern. We apologize for any confusion in not clearly explaining
> the results and connecting them to practical scenarios. Building upon the details provided
> in our response to your first weakness listed, the observed gap can also be ascribed to the
> potential simplicity of the baseline generating mechanism, possibly being a linear model or closely
> resembling one. In such cases, the perceived benefits may not seem to be substantial compared to
> other competing algorithms. Nevertheless, the significant increase in cumulative reward indicates
> the robustness of our algorithm and its superior performance in unknown environments

---

### Author Response · Authors · 2023-11-23
**General Response to Reviewers**

We thank the reviewers for their helpful comments. We have revised the paper based on them. We would like to point out the following areas where we have made substantial changes.
* We added a new introductory paragraph giving more background on why bandits are important in mHealth.
* We added some discussion of doubly robust bandits in the related work.
* We improved on the assumptions required (assumption 3) on the convergence of the mean function estimate to the true mean function. This assumption (L2 convergence in little o rather than uniform convergence in big O) is weaker and matches the standard assumption used in the DML literature (e.g. Chen et al. 2022)
* We added additional discussion of the regret bound interpretation and challenges afters its statement.

In addition to these changes, we address individual reviewer comments and questions below.

---

### Meta-Review · Area_Chair_KXFB · 2023-12-08

**Metareview:**

This paper proposes a contextual bandit algorithm for personalized mobile health (mHealth) applications (e.g., prompting a user to get up and go for a walk). The algorithm is based on Thompson sampling, with two extensions: (1) _nearest-neighbor regularization_ (NNR), such that the parameters of connected users (in an observed user and time graphs) should be closer; (2) _double machine learning_ (DML) to model baseline rewards. The paper provides a regret guarantee for the proposed algorithm, and conducts experiments on both simulated data and 2 real mHealth datasets, which shows superiority of the proposed method over some comparable baselines.

The reviewers agreed that mHealth is an important topic, and that the paper did a good job of surveying prior work in this area. They appreciated the rigorous analysis of the proposed algorithm, and none cited any issues with the theory.

However, a common complaint was that the paper's writing was confusing. It lacks proper motivation for the proposed technique; it lacks sufficient intuition for the many assumptions needed for the theory, and explanations of why they are reasonable assumptions. I agree with the reviewers on this point. For instance, I was confused as to where the user and time graphs come from -- are they observed _a priori_, or latent? Some reviewers were also unsure how much of the proposed approach was novel compared to prior work. Other perceived weaknesses -- such as marginal improvement over baseline, or lack of real data experiments -- I believe can be chalked up to misunderstandings.

The authors' rebuttal and revision may have clarified some ambiguities -- in particular, regarding the novelty of the proposed approach with respect to prior work. However, I must note that the rebuttal came at the very end of the author-reviewer discussion period, thereby preventing any iterative interaction between the authors and reviewers. This is a shame, because some reviewers had unresolved questions/concerns.

My impression is that there is some solid work here, but the presentation really needs to be improved and made more accessible. Sometimes this can be addressed in a camera-ready revision; but in this case, I think there is just too much that needs to be updated. I therefore cannot recommend acceptance, but I urge the authors to revise the paper -- incorporating the feedback here -- and resubmit it.

**Justification For Why Not Higher Score:**

The paper needs a major rewrite, and it should be reviewed post-rewrite. Plus, the consistently borderline scores indicate a general lack of interest in this work (which might be addressed by rewriting the motivation), so it's hard for me to justify overriding their scores.

**Justification For Why Not Lower Score:**

N/A

---

### Decision · Program_Chairs · 2024-01-16

Reject